# PREDICT: Preference Reasoning by Evaluating Decomposed preferences Inferred from Candidate Trajectories

## Abstract

Accommodating human preferences is essential for creating AI agents that deliver personalized and effective interactions. Recent work has shown the potential for LLMs to infer preferences from user interactions, but they often produce broad and generic preferences, failing to capture the unique and individualized nature of human preferences. This paper introduces PREDICT, a method designed to enhance the precision and adaptability of inferring preferences. PREDICT incorporates three key elements: (1) iterative refinement of inferred preferences, (2) decomposition of preferences into constituent components, and (3) validation of preferences across multiple trajectories. We evaluate PREDICT on two distinct environments: a gridworld setting and a new text-domain environment (PLUME). PREDICT more accurately infers nuanced human preferences improving over existing baselines by 66.2% (gridworld environment) and 41.0% (PLUME).

## 1    Introduction

A fundamental component of effective interaction is understanding the preferences of those with whom we engage. Successfully recognizing and accommodating these preferences leads to more pleasant and efficient experiences (Felfernig et al., 2006). While such preferences can be verbalized explicitly, they can also be inferred implicitly from past interactions. Ideally, an AI agent should be able to both use explicit feedback and learn from implicit cues. As human directions are commonly expressed in natural language, creating a mapping from implicit cues to natural language could enable a natural-language conditioned agent to seamlessly integrate both implicitly and explicitly defined preferences. This work focuses on this gap by proposing a method to infer natural language preferences from a user's actions.

Large Language Models (LLMs) possess strong priors about human behavior (Brown et al., 2020). Previous work has demonstrated that these priors can provide the basis to infer user preferences in domains such as robotic manipulation (Peng et al., 2024) and collaborative authoring (Gao et al., 2024). However, current methods infer preferences without reflection nor refinement, resulting in generic outcomes that limit the models' adaptability toward the uniqueness and nuance of an individual's preferences. We propose **PREDICT** (**P**reference **R**easoning by **E**valuating **D**ecomposed preferences **I**nferred from **C**ounterfactual **T**rajectories), which is **comprised of three algorithmic contributions** to enhance the precision and efficiency of preference inference: (1) iteratively refining inferred preferences until the induced trajectory closely aligns with the user's example, (2) breaking down (or decomposing) inferred preferences into constituent components, and (3) validating the inferred preferences across multiple user examples. The preference inferred by predict are used to condition the behavior or generations of an AI assistant.

We systematically demonstrate the benefits of PREDICT's contributions on two environments: a gridworld environment, where an agent learns to pick up objects based on a user's preferences over colors and shapes, and PLUME, a text-based environment where an agent learns to write text that aligns with a user's preferences. PLUME is a new environment and is **a contribution of this paper**. PREDICT demonstrates improvements of 66.2% over behavioral cloning in the gridworld environment, and 41.0% over CIPHER (Gao et al., 2024) in PLUME. In PLUME, we augment PREDICT with in-context learning and achieve a further 17.9% improvement. The innovations

outlined in this paper are a key step toward more personalized and effective interactions between AI agents and humans.

## 2 RELATED WORK

**Natural Language Conditioned Agents** Language is the most natural way for humans to communicate and express themselves. As a result, considerable research has focused on natural language-conditioned agents across a variety of domains. BabyAI (Chevalier-Boisvert et al., 2019) introduces an environment for natural language conditioned gridworld agents. Further advancements, such as gSCAN (Qiu et al., 2021), investigate how gridworld agents handle compositionality, while Zhong et al. (2020) explore the ability of agents to learn environment dynamics from text. Misra et al. (2018) proposes LingUNet as way to fuse language and vision in a simulated 3D world. Blukis et al. (2018; 2020) extend this for continuous drone control. In room-to-room navigation, works such as CLIPNav (Du et al., 2023) and Embodied CLIP (Khandelwal et al., 2022) use CLIP embeddings (Radford et al., 2021) to condition agents on visual-language aligned representations.

In robotic arm manipulation, Lynch & Sermanet (2021) condition trajectories on both goal images and natural language, demonstrating successful task completion with limited language labelling. Jang et al. (2021) builds upon this and use videos as goal contexts. For pick-and-place tasks, Shridhar et al. (2022) uses a CLIP-based two-stream architecture, and Mees et al. (2022; 2023) demonstrate long-horizon task completion via hierarchical approaches.

In natural language generation, prompting (Radford et al., 2019) and in-context learning (Brown et al., 2020) have proven effective methods for controlling the generation of text, especially in a preference-driven context (Sun et al., 2023; 2024).

**Personalization** Some prior approaches of adapting models to user preferences involve RLHF (Stiennon et al., 2020) and fine-tuning (Tan et al., 2024; Zhuang et al., 2024), which can be compute-intensive and inaccessible to some practitioners without the budget or scale of needed data. With the rise of LLMs with strong instruction-following capabilities, methods like prompting to adapt a user's profile have become more popular (Shen et al., 2024; Salemi et al., 2024), however these approaches often rely on explicit feedback provided from the user (Lin et al., 2024). PREDICT circumvents these issues by learning from implicit user signals, breaking down preferences into sub-components to generate tailored user-preferences, all without the need of fine-tuning.

**Preference-Conditioned Agents** Combined preference inference and conditioning has recently gained traction, with the following two works being most aligned with our approach.

Peng et al. (2024) explores preference learning in quadrupedal mobile manipulation using an object detection module to map image observations to text. An LLM then infers preferences by comparing pairs of trajectories. These preferences are in turn used to improve task alignment with user preferences. Gao et al. (2024) propose the PRELUDE environment, where an LLM learns writing style preferences in a collaborative authoring task. We discuss this work in detail in Section 4.3.

These methods rely on a single inference step, whereas our approach uses iterative refinement for more precise preferences, and validation across several user examples for robustness.

## 3 PREDICT

We now outline PREDICT's key contributions to preference inference. Whenever a user provides a demonstration for how to complete a task the user would like their AI assistant to be able to complete, PREDICT improves the inferred preferences using: (1) iterative refinement and (2) preference validation against relevant user examples. Iterative refinement consists of two sub-steps: (i) update inferred preferences through candidate trajectories, and (ii) breaking down the inferred preferences into constituent components. Iterative refinement is halted when either the maximum number of iterative refinement steps is reached or no updates are made in sub-step (i). The inferred preferences are then used to condition and align the behaviors or generations of an AI assistant. A visualization of PREDICT along with summaries of the prompts used for each of the steps above are provided

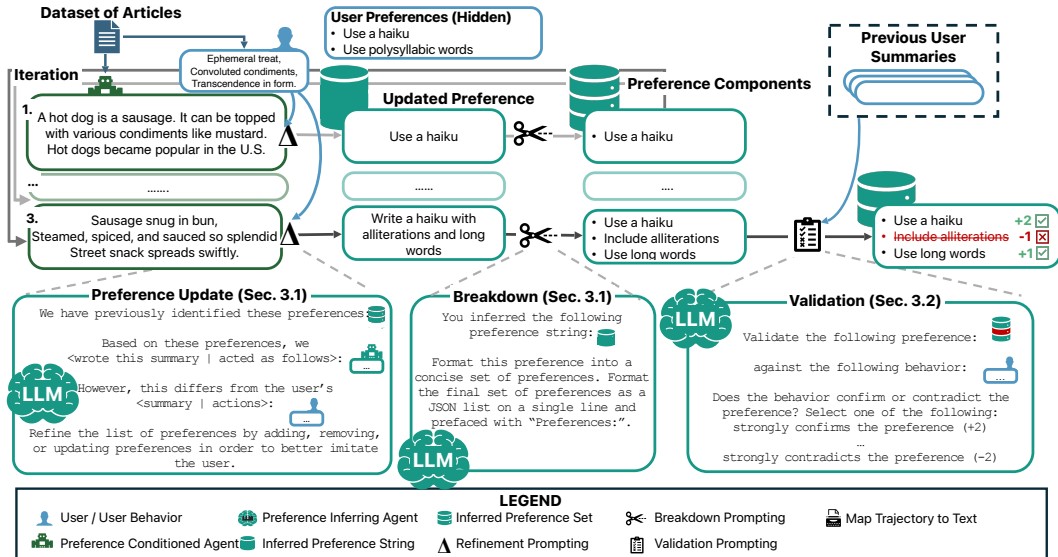

Figure 1: Overview of PREDICT on PLUME's summary writing task. The user provides a writing demonstration, which PREDICT can learn from. PREDICT observes the demonstration, then executes an iterative refinement (preference update and breakdown) and a validation step. Iterative refinement updates the set of inferred references by generating a new candidate solution using the currently inferred preference set then prompting an LLM to compare the candidate solution to the user's demonstration and update the preference set to more closely match the user's writing. An LLM is then prompted to break it into a set of component parts. Iterative refinement continues until a candidate solution matches the demonstration or a maximum number of iterations is reached. Once the preferences are updated, each preference component is validated with LLM-as-a-Judge to evaluate how well the component aligns with other user demonstrations.

in Fig. 1, and the algorithm is provided in Appendix C. The complete prompts are shown in Appendix H.1 (Fig. 8 and 9)[1].

## 3.1 ITERATIVE REFINEMENT

PREDICT conditions the AI assistant (e.g. an LLM) on the inferred user preferences to generate a candidate solution (e.g. summary, email, or trajectory) for a given task (e.g. summarize an article). Example candidate solutions (e.g. haikus on the top and trajectories on the bottom) are provided in Fig. 1 on the far left. If no prior user demonstrations have been seen, the AI assistant is conditioned on an empty preference set.

The candidate solution is then compared to the user's demonstration. If the candidate trajectory exactly matches the user's demonstration then the current inferred preferences are considered sufficient to explain the user's behavior and no further learning is required. This means all subsequent steps in the PREDICT algorithm are skipped.

However, if the assistant's solution differs, we prompt an LLM (prompt outline: Fig. 1 "Iterative Refinement" [update here and in figure to "Preference Update"]) to **update the inferred preferences** so that they explain the difference between the candidate solution and the user's demonstration. Examples of inferred preferences can be seen in Fig. 1 "Updated Preferences" [need to add this label to figure].

PREDICT then **breaks down the updated preferences** by prompting an LLM to break the updated preferences into their components parts (prompt outline: Fig. 1 "Breakdown"). Examples of LLM-identified preference components are provided in Fig. 1 "Preference Components". Breaking down the preferences provides several advantages. The components provide greater coverage of the preference space with less data, e.g., three components can be combined to cover nine distinct

---

[1]code coming soon!

preference sets. Further, preference components make it easier to refine preference sets by adding, removing, or modifying single components rather than modifying long-form preference descriptions (see Fig. 1 "Updated Preferences" for examples). Lastly, preference components remove ambiguity when validating preferences. If we validate a compound preference and only a single component is incorrect, then the entire compound preference including any useful components may be discarded.

PREDICT then generates a new candidate trajectory by conditioning the AI assistant on the updated preference components. We repeat the process of generating AI assistant solutions, comparing to the user demonstrations, updating the inferred preferences, and breaking the updated preferences into components until the candidate solutions exactly match the user's demonstrations, or a maximum number of iteration steps is reached. In all experiments, we use a maximum of three preference update steps per user demonstration.

### 3.2 Validating Preferences

PREDICT validates each preference component in the inferred preference set against each of the most relevant user demonstrations by prompting an LLM to determine whether the demonstration strongly confirms, somewhat confirms, is neutral toward, somewhat contradicts, or strongly contradicts the preference (prompt outline: Fig. 1 "Validation"). Each answer is mapped to a score from +2 (strongly confirms) to -2 (strongly contradicts). If the mean score across all demonstrations is below a manually specified threshold, the preference is removed. Examples of retained and discarded preference components are in Fig. 1 "Validated Preferences". To avoid discarding correct, but rare, preferences, a preference component must be validated against a minimum of two user demonstrations before it can be removed. In our experiments, we use a validation threshold of 0.25.

**Preference Aggregation** Following CIPHER Gao et al. (2024), before solving a new task, PREDICT retrieves up to five previous, relevant examples. The preferences inferred for each example are aggregated, and an LLM is prompted to remove redundancy and condense the combined set of preferences. The condensed preference set is then used to complete the task.

## 4 Experimental Set Up

All of our experiments consist of three phases per task. First, the user completes the task using their true preferences. Second, the agent attempts to complete the task using its currently inferred preferences (if any). Finally, the agent compares its attempt at task completion with the user's example to infer new preferences to use going forward.

All agents are evaluated along two dimensions: *preference quality* that measures similarity between the true and inferred preference sets, and *action quality* that evaluates an agent's task completion against the user's true preferences. Note that the first task completion will always be conditioned on an empty preference set and that we evaluate the preference set used to solve the task. Thus, the first step is equivalent across all agents, and we omit its results.

The agent learns from $4-10$ users (depending on the task) with five examples per user, and performance is reported as the mean across all examples, users, and across five seeds (standard deviation is reported over these seeds). The user preferences for the assistive writing tasks are in Appendix F (Table 5), whereas the PICK UP task has a rule-based user preference construction procedure described in Section 4.2. For all experiments, we use GPT-4o as the inferring agent except when we compare LLMs of different sizes and quality (shown in Fig. 2) [2]. For the assistive writing tasks, GPT-4o is used as a synthetic human. The synthetic human prompts can be found in Appendix H.2.

### 4.1 Research Questions

We pose the following research questions:

**RQ1: Does iteratively creating candidate comparison trajectories improve the quality of inferred preferences?** To explore this, we consider three variants of PREDICT: (1) $\text{PREDICT}_{1NC}$ (1NC=1 inference step, no candidate) uses no example comparisons and prompts the LLM a single

---

[2]Determined by MMLU performance: llama8b (68.4) and 70b (82); GPT-4o-mini (82) and GPT-4o (88.7)

step to infer the preference given only the user's example; $\text{PREDICT}_{\text{1SC}}$ (1SC=1 inference step, single candidate) is PREDICT with a single inference step and a single candidate trajectory; and (3) $\text{PREDICT}_{\text{SC}}$ ($\leq$3 inference steps, single candidate) is PREDICT with a single candidate used for all inference steps. Comparing $\text{PREDICT}_{\text{1NC}}$ and $\text{PREDICT}_{\text{SC}}$ measures the effect of comparing candidate examples to the user's examples when inferring preferences. The differences between the $\text{PREDICT}_{\text{1SC}}$ and $\text{PREDICT}_{\text{SC}}$ variants quantifies the role of increasing the number of inference steps, while comparing $\text{PREDICT}_{\text{SC}}$ and the full PREDICT algorithm clarifies the effects of explicitly providing the LLM with the outcomes of its predictions.

**RQ2: Does breaking down preferences into components improve the performance and consistency of the preference inferring methods?** To answer this question we compare the full PREDICT algorithm with a variant that does not breakdown preferences $\text{PREDICT}_{\text{CP}}$ (CP=compound preferences). We hypothesize that PREDICT improves performance and reduces variance between seeds relative to $\text{PREDICT}_{\text{CP}}$.

**RQ3: Does filtering preferences by validating them across multiple examples lead to fewer errors?** To answer this question, we evaluate a variant, $\text{PREDICT}_{\text{NV}}$ (NV=no validation), that does not validate preferences.

## 4.2 ENVIRONMENT 1: PICK UP

We develop the PICK UP (**P**olicy **I**mitation by **C**omprehending **K**ey **U**ser **P**references) task in a grid-world environment populated with various objects of different shapes and colors. See Appendix A Fig. 4 for an overview of PREDICT applied to the PICK UP task. Users in the environment navigate to pick up objects with attributes (i.e., shape/color) they like, while avoiding objects with attributes they dislike, before navigating to an end goal location. When an object is collected, a reward of +1 is awarded for each liked attribute and a reward of -1 is awarded for each disliked attribute. For example, an object whose shape and color are both liked would have a reward of +2, while an object whose shape is liked and color disliked has a reward of 0. Note the reward function is used only for evaluation purposes and does not play a role in preference learning.

For PICK UP, we automatically transform trajectories into a structured language description. Fig. 5 (Appendix D) shows a visual and natural language representation of the environment and its trajectories. The objective of a preference inferring agent in this environment is to be able to collect the same objects that the user's would. To accomplish this, they must first identify the user's likes and dislikes, and then navigate the world to collect the appropriate objects. We include the presence of neutral objects in the environment, which adds ambiguity to the system as neutral objects are only picked up if they are along the shortest path between desirable objects or the goal, which is not identifiable from the text representation of the user's example. Thus, from the perspective of an inferring LLM, *the environment is only partially observable*. This design is intentional; motion is inherently difficult to encode in language, so many tasks will be partially observable to an LLM. Due to the partial observability, we require three validations to discard a preference in PICK UP.

In this environment, each task instance is defined by a user identifier and an environment layout containing seven random objects placed at random locations in a 5x5 grid. The user identifier maps to a unique and private set of preferences. Each user's preference set contains exactly one liked shape, one liked color, one disliked shape, and one disliked color, however this information is not provided to the inferring agent. These are all specified in the structured format: `<likes/dislikes><attribute>`. Users are neutral toward all the remaining attributes. The well-defined structure of the preferences in PICK UP allows us to map a preference set to a set of positive reward objects and negative reward objects. We then use this mapping to condition an A* agent that collects all the positive objects while avoiding negative objects.

The preference structure also enables direct comparison of preferences. To this end, we report the Intersection over Union (IoU) between the inferred and true preference sets as the *preference quality metric*. A downside of the rigid preference structure is that it requires us to decompose preferences, which prevents us from addressing RQ2 in this environment. For the *action quality metric*, we measure the cumulative reward, or return, of the agent's trajectories. Each liked/disliked attribute (shape or color) in the set of collected objects adds +1/-1 to the score respectively. For all experiments, we use 10 distinct users (N = 10).

### 4.3 Environment 2: Assistive Writing

**PRELUDE:** Gao et al. (2024) propose PRELUDE (PREference Learning from User's Direct Edits) as an environment to evaluate preference inferring algorithms. PRELUDE consists of two tasks: summarizing articles and writing emails from notes. Each task has a set of users with each user having a distinct set of preferences. Each user additionally writes their summaries/emails on different topics, with each topic corresponding to a different source of articles/notes (e.g., chat forums, paper abstracts, encyclopedia articles). The summarization and email writing tasks have five and four users respectively.

For each task instance, the agent must write a summary or email using the article / notes and any inferred preferences it has learned up to that point. The user is then asked if the agent's output is satisfactory based on their true preferences. If the agent's output is satisfactory, the cost to the agent is zero. If the agent's output is not satisfactory, the user edits the agent's output according to their preferences, and a cost based on the extent of the edit is incurred.

**PLUME:** The objective of the PRELUDE environments is to evaluate how well a model infers a user's preferences and the cost of incorrectly inferred preferences. Therefore, it is vital that the measure of inferred preference quality is highly correlated with the cost function. We analyze PRELUDE (see below) and find that the chosen metrics, the editing process, and the sets of preferences used are key limitations of the environment, which contribute to a weak correlation between the quality of the preferences and the quality of the generated writing.

For these reasons, we develop a new environment based on same underlying tasks as PRELUDE, which we call **PLUME**: Preference Learning from User Memos and Emails. As in Gao et al. (2024), PLUME uses GPT-4o as a proxy-human to be our user. In the following sections, we provide a detailed description of each limitation and how it is addressed by PLUME. An example of how PREDICT is applied to PLUME's summary task can be seen in Fig. 1.

**Metric Correlation** We begin by investigating the magnitude of the correlation between the proposed *preference quality metric* — preference set accuracy[3] — and *action quality metric* — Levenshtein distance (Levenshtein, 1966) — used in PRELUDE (Gao et al., 2024). To find this correlation, for a given context, we generate the powerset of the preference set. We then create a population of agents, each conditioned on one of the subsets from the powerset. These agents and a user complete five instances of the task within their context, on which we measure the preference and action quality. Intuitively, agents conditioned on larger subsets of the true preference set have a higher preference quality score and their generation quality should reflect this. We repeat across every context and both tasks, and calculate the Pearson correlation between every *preference quality metric* and every *action quality metric*. The results are shown in Appendix E.3 Table 4 (for metric correlation by task).

The results, reported in the first column of Appendix E.3, show a weak correlation ($< 0.5$) between the PRELUDE's preference accuracy and Levenshtein distance. This can be explained by the inherent limitations with the metrics. The accuracy metric relies on the "highest" BERTScore, and therefore cannot differentiate partially correct preferences from perfectly correct preferences. Moreover, the Levenshtein distance can vary substantially between generations, leading to a wide range of possible costs even when the exact same preferences are used for the generation (an illustrative example of this is shown in Appendix G.1). Gao et al. (2024) allude to this fact as a motivation for their two-stage editing process, and when we compare the results to a version of PRELUDE where the user always generates summaries/emails directly from the article/notes instead of editing the agent's summary/email (PRELUDE$_{\text{NoEdit}}$), we see a further drop in correlation. However, we propose addressing this issue using improved metrics, as the editing procedure itself imposes notable limitations, which are discussed below.

To this end, we investigate and compare several new preference and generation-quality metrics. For the *preference quality metric*, we test using the BERTScore (Zhang* et al., 2020) directly. For the *action quality metric*, we additionally test length-normalized Levenshtein distance (ln-L-dist), BERTScore, and an LLM-as-a-Judge (Zheng et al., 2023) metric inspired from the editing procedure in PRELUDE. The LLM-as-a-Judge evaluation is a per preference-component match (PPCM) that

---

[3]a preference is correct if its BERTScore (Zhang* et al., 2020) with true preference set is greater than the BERTScore with any other preference set.

asks an LLM how much a component of a preference is exhibited in a piece of writing on a five point scale from "clearly contradicts" (score of -2) to "clearly exhibits" (score of +2). This is repeated for each component of the true preference set, and we compute the mean score across components. The full prompts used for both of these metrics are shown in Appendix H.4 (Fig. 13).

The results in Table 4 (Appendix E.3) show that BERTScore has a stronger correlation than PRE-LUDE's accuracy metric with every writing generation metric compared. Looking at action/generation quality metrics, Levenshtein distance consistently has the weakest correlation, while PPCM has the strongest. Notably, the pairing of BERTScore (preference quality) and PPCM (generation quality) provides the highest correlation in every situation and are the primary metrics we report in PLUME.

**The Editing Procedure** Asking whether a generation matches the user's preferences is inherently ambiguous in cases where they only partially meet the user's preferences. Even if this ambiguity is resolved, generations that are not selected for editing incur no cost, which removes any incentive to further improve the quality of the learned preferences. This limits the environments ability to differentiate a wide range of methods. Lastly, the editing process unduly influences the user's writing, as demonstrated in Appendix G.2.

In place of the editing, PLUME has the agent and user independently solve the task at every step. This removes any ambiguity on whether a generation should be edited and incur a cost, provides a smoother curve along which to evaluate different methods, prevents agents from influencing users, and enables the agent to learn from every user example.

**Preference Sets** We observe the following limitations with PRELUDE's preference sets: (1) certain preference components have little impact on the generated text, due to unclear definitions (e.g., `skillful foreshadowing`) or similarity to default LLM behavior (e.g., `clear`); (2) Some preferences are repeated across several contexts (e.g., `short`, `brief`, `concise` appear in four of five summarization contexts); and (3) There is a large variance in preference set complexities (e.g., `targeted to young children`, `storytelling`, `short sentences`, `playful language`, `interactive`, `positive` vs. `question answering style`).

To address these, PLUME reworks the preference sets with the following criteria: (1) each preference set contains an equal number of components, (2) within each task, preference sets should have a shared structure, (3) as much as possible, preferences should be orthogonal to each other, avoiding overlapping preferences (e.g., `write in the style of old-timey radio` and `use archaic language`) or contradictory preferences (e.g., `use emojis` and `use a formal tone`). (4) Preferences should not follow an LLMs default biases — i.e., generating an output conditioned on no preference should lead to a low score. A full list of the preferences used in PRELUDE and PLUME is shown in Appendix F (Table 5). We encourage future researchers to use PLUME with different preference sets to adjust difficulty or examine specific concepts.

**Knowledge of Contexts** Instead of treating each article/notes topic as a distinct user, PRELUDE introduces the additional challenge of context awareness where a single user has different preferences based on the topic of the article/notes. Therefore, prior to writing a summary or an email the agent must first identify the correct context. However, this is orthogonal to the challenge of inferring preferences from user examples. As this work focuses on how to infer preferences, the version of PLUME used in all experiments assume a distinct known user per topic. We note that PLUME is easily adaptable to use hidden contexts if desired.

## 4.4 BASELINES

In addition to the PREDICT baselines outlined Section 4.1, we implement the following models.

In PICK UP, we implement behavioral cloning (BC) (Pomerleau, 1988) that is trained using a cross-entropy loss on the related user examples seen to date. Due to the low-data regime, we first pre-train the BC agent on a dataset (1000 trajectories, $\sim 12,000$ state-action pairs) of distinct user examples whose preference sets differ from those found in the gridworld environment. During evaluation, when the BC agent sees a new user example, it adds the example to its dataset of user specific examples. It then creates a clone of its pre-trained agent and fine-tunes a version of the agent on examples from the same user, early stopping on a single user example reserved for validation.

| Method | PICK UP | | Summarization | | Emails | |
|---|---|---|---|---|---|---|
| | IoU | Return | BScore | PPCM | BScore | PPCM |
| No Learning Baselines | | | | | | |
| NP | $0.00_{\pm 0.00}$ | $-0.07_{\pm 0.03}$ | $-0.50_{\pm 0.00}$ | $-1.49_{\pm 0.15}$ | $-0.50_{\pm 0.00}$ | $-1.07_{\pm 0.17}$ |
| Oracle | $1.00_{\pm 0.00}$ | $2.06_{\pm 0.19}$ | $1.00_{\pm 0.00}$ | $1.68_{\pm 0.07}$ | $1.00_{\pm 0.00}$ | $1.84_{\pm 0.04}$ |
| Learning Baselines | | | | | | |
| BC | $0.00_{\pm 0.00}$ | $-0.01_{\pm 0.10}$ | - | - | - | - |
| ICL | - | - | $-0.50_{\pm 0.00}$ | $1.07_{\pm 0.22}$ | $-0.50_{\pm 0.00}$ | $1.11_{\pm 0.17}$ |
| C1 | - | - | $0.12_{\pm 0.01}$ | $-0.58_{\pm 0.12}$ | $0.12_{\pm 0.01}$ | $-0.04_{\pm 0.20}$ |
| C5 | - | - | $0.07_{\pm 0.01}$ | $-0.66_{\pm 0.04}$ | $0.07_{\pm 0.02}$ | $-0.14_{\pm 0.12}$ |
| PREDICT Ablations | | | | | | |
| Base | $0.41_{\pm 0.07}$ | $1.22_{\pm 0.15}$ | $0.18_{\pm 0.02}$ | $0.38_{\pm 0.14}$ | $0.15_{\pm 0.02}$ | $0.90_{\pm 0.16}$ |
| 1NC | $0.42_{\pm 0.04}$ | $1.24_{\pm 0.28}$ | $0.17_{\pm 0.02}$ | $0.25_{\pm 0.09}$ | $0.16_{\pm 0.02}$ | $1.02_{\pm 0.34}$ |
| 1SC | $0.43_{\pm 0.07}$ | $1.18_{\pm 0.20}$ | $\mathbf{0.29_{\pm 0.01}}$ | $0.49_{\pm 0.13}$ | $\mathbf{0.24_{\pm 0.05}}$ | $0.95_{\pm 0.12}$ |
| SC | $0.45_{\pm 0.02}$ | $1.24_{\pm 0.27}$ | $0.25_{\pm 0.01}$ | $0.68_{\pm 0.17}$ | $0.22_{\pm 0.02}$ | $1.07_{\pm 0.27}$ |
| CP | - | - | $0.15_{\pm 0.02}$ | $0.81_{\pm 0.13}$ | $0.13_{\pm 0.02}$ | $1.09_{\pm 0.28}$ |
| NV | $0.48_{\pm 0.06}$ | $1.25_{\pm 0.17}$ | $0.26_{\pm 0.03}$ | $0.73_{\pm 0.18}$ | $0.20_{\pm 0.01}$ | $0.95_{\pm 0.15}$ |
| Full | $\mathbf{0.49_{\pm 0.06}}$ | $\mathbf{1.40_{\pm 0.15}}$ | $0.27_{\pm 0.03}$ | $0.78_{\pm 0.06}$ | $0.23_{\pm 0.02}$ | $1.10_{\pm 0.10}$ |
| +ICL | - | - | $0.26_{\pm 0.02}$ | $\mathbf{1.32_{\pm 0.20}}$ | $0.20_{\pm 0.02}$ | $\mathbf{1.64_{\pm 0.14}}$ |

Table 1: **PREDICT Iterative Refinement Steps = 3** Main Results. PREDICT's ability to infer the correct preference set and quality of generated behaviors. Results are reported as the mean and standard deviation across five seeds. For all metrics, a higher score is better. Acronym Glossary: IoU (Intersection over Union), BScore (BERTScore), PPCM (per preference-component match), NP (No-Preferences), BC (behavioral cloning), ICL (in-context learning), C1/C5 (CIPHER-1/5), 1NC (1-step No Candidate), 1SC (1-step Single Candidate), SC (Single Candidate), CP (Compound Preferences), NV (No Validation).

In PLUME, we implement CIPHER-1 and CIPHER-5 (Gao et al., 2024), and an in-context learning (ICL) agent using previously observed article/notes and resulting user summary/email as examples.

We then implement three additional baselines across both environments. An agent that solves the task with no preferences (NP), providing a lower-bound of performance. An oracle agent that receives access to the user's true preference, providing an upper bound of performance, and PREDICT$_{\text{Base}}$, which is a variation of PREDICT that uses only a single candidate trajectory, a single inference step, compound preferences instead of a set of preference components, and uses no validation. We note that PREDICT$_{\text{Base}}$ is conceptually equivalent to CIPHER, however it uses the prompts from PREDICT, which differ from those in CIPHER.

## 5 RESULTS AND DISCUSSION

We present our main results comparing baselines and various PREDICT ablations in Table 1. Results on PRELUDE can be found in Appendix E.2. To compare tasks on action quality with metrics on different scales, we use a percentile score, where 0% corresponds to the no-preference (NP) baseline and 100% to the oracle preference baseline. All percentage improvements are reported as the difference in scores on this scale. Overall, PREDICT$_{\text{Full}}$ outperforms PREDICT$_{\text{Base}}$ by 9.3%, BC by 66.2%, and CIPHER by 41.0%.

**RQ1.** In our first question, we set out to verify whether generating iterative candidate trajectories is beneficial to inferring preferences. Comparing PREDICT to its ablated versions on the action/generation quality metric (PPCM), shows each component of the iterative refinement process improves performance. Comparing PREDICT with no comparison trajectory — PREDICT$_{\text{1NC}}$ — to PREDICT with a single candidate comparison trajectory — PREDICT$_{\text{1SC}}$ — we can see providing comparison trajectories is beneficial when inferring preferences (2.3% mean improvement). This result supports the algorithmic decisions in (Gao et al., 2024; Peng et al., 2024). Allowing for multiple refinement steps provides a further increase in performance (Table 1: PREDICT$_{\text{1SC}}$ vs. PREDICT$_{\text{SC}}$,

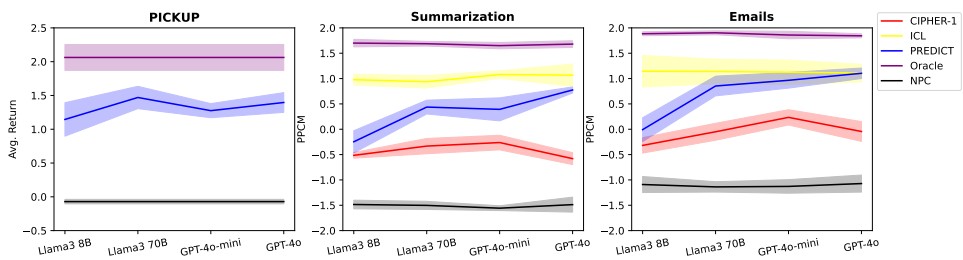

Figure 2: Mean and standard deviation (5 seeds) performance for CIPHER-1, in-context learning (ICL), PREDICT, Oracle, and no preferences (NPC) for different preference-inferring LLMs.

4.3% mean improvement). This can be explained by the LLM having more chances to infer correct preferences. Lastly, when comparing PREDICT$_{SC}$ to PREDICT$_{Full}$ we see another 3.9% improvement. This highlights the benefits of updating candidates after each inference step using the newly inferred preferences. In all, iterative refinement provides a mean improvement of 9.0%.

**RQ2.** We next examine the effect of splitting compound preferences down into their constituent components by comparing PREDICT to PREDICT$_{CP}$ (compound preferences) (Table 1) on PLUME. Action/generation quality (PPCM) does not show a clear trend, with both models achieving similar scores on both tasks. However the full version of PREDICT does achieve a higher preference BERTScore on both tasks; in fact PREDICT$_{CP}$ produces one of the lowest BERTScores. More interesting however, is the variance: using compound preferences leads to high PPCM variance, whereas the full version of PREDICT is the most consistent performer (lowest variance). We hypothesize that splitting preferences into components enforces structure and benefits consistency, but it also prevents the LLM from using the more complex, multi-faceted preferences that PREDICT$_{CP}$ can utilize. On the other hand, when PREDICT$_{CP}$ makes an error, the error is much more difficult to isolate and rectify. This can lead to PREDICT$_{CP}$ retaining incorrect preferences or discarding everything. More work is required to investigate and potentially mitigate this trade-off.

**RQ3.** We investigate the benefit of validating preferences by comparing PREDICT to PREDICT$_{NV}$ (no validation). Here, we see a modest but consistent action quality benefit of 7.0%, 1.6%, and 5.2% for the PICK UP, email writing, and summarization tasks respectively when using validation.

**Discussion.** Fig. 2 shows that the performance of PREDICT scales better with the quality of the underlying LLM (e.g., Llama3 70B-instruct vs. GPT-4o), compared to every other methods. Additionally, as expected, performance increases as more user examples are seen (Fig. 6), with the largest performance gain from the first example.

While BERTScore is a more representative metric than the accuracy used in Gao et al. (2024), it does not fully capture the impact of the inferred preferences: preferences can be written very differently, but lead to similar outcomes (e.g., `use hashtags for emphasis` versus `write in the style of tweet`). For this reason, we focus primarily on action quality in this paper, but encourage future work to investigate alternatives metrics that better capture preference intent.

While PREDICT outperforms all preference-conditioned and no-learning baselines, ICL and PREDICT perform equally well on email writing with ICL outperforming PREDICT on summarization. All summarization tasks have a formatting/structure preference (e.g., `write in the style of a tweet`), which are difficult to capture using natural language preference descriptions. PREDICT often tries to capture these preferences using multiple relevant, but imperfect preferences (e.g., `use hashtags for emphasis`, `include emojis to create a playful tone`, `employ attention grabbing phrasing`). We further investigate the performance gap by comparing the performance across preference sets (Fig. 3) and find that ICL excels on sets with the strongest structural preferences (e.g., `write in the style of a screenplay`). In contrast, PREDICT outperforms ICL on the preference sets requiring a more nuanced understanding of tone (e.g., `be intensely emotional` or `be sharply critical`). While ICL generally performs well, preference conditioning has several advantages: (1) preferences are easier to interact with than a dataset of in-context examples, (2) at inference time, it requires 10x fewer tokens, and (3) it can benefit a wider range of tasks, e.g., human-agent

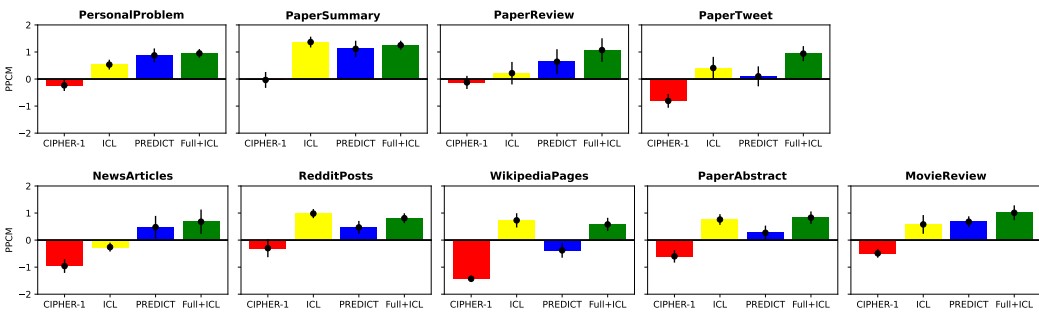

Figure 3: PPCM mean and standard deviation (5 seeds) for PREDICT, CIPHER-1, and in-context learning (ICL) by **Email** (top) and **Summary** (bottom) sub-task type. GPT-4o is the LLM.

collaboration (Liu et al., 2024), sample efficient imitation/reinforcement learning, or generating personalized preference pairs for RLAIF (Sun et al., 2024).

As the two methods seem to have complementary benefits, we combine the two methods (PREDICT$_{\text{Full+ICL}}$), and achieve a performance gain of 17.9% and 13.1% over PREDICT and ICL respectively. **PREDICT$_{\text{Full+ICL}}$ outperforms previous state-of-the-art CIPHER by 58.8%.**

### 5.1 LIMITATIONS AND FUTURE WORK

While the methods proposed in this work provide a number of significant improvements, their limitations and challenges provide interesting avenues for future work.

First, in this paper we focus on learning with the fewest user examples possible. However another aspect of efficiency is the total number of prompt and generated tokens used, and adding more refinement and preference validation steps increases the number of tokens used. In our experiments, PREDICT$_{\text{Full}}$ used (5.87x / 6.07x) more (prompt / generated) tokens on average than PREDICT$_{\text{Base}}$. Given the monetary and environmental cost of LLMs, reducing the number of tokens while retaining performance is an important area for improvement.

Another limitation is the requirement to represent all trajectories in language. While this is possible for the environments used here, it may not be possible in all domains (e.g., any environment requiring an understanding of subtle movement patterns). Future work is needed to investigate the use of multimodal foundations models, such as VLMs, to address this limitation.

Lastly, a full-scale human trial would provide a greater understanding of the benefits and limitations of the proposed method. We look forward to investigating this more closely in future work.

**Ethical Concerns** The proposed method allows for greater personalization of assistive agents. However inferring a user's preferences could be seen as an invasion of privacy. With this in mind, these methods should be applied only with explicit consent from human users.

## 6 CONCLUSION

In this paper, we propose three novel contributions to guide an LLM to better infer preferences from user examples and introduce a new environment for evaluation. First, we iteratively refine preferences by using a preference conditioned agent to test inferred preferences. Second, we break preferences down into their constituent components. Third, we validate preferences against other user examples. We demonstrate on both navigation and writing environments that the proposed method improves performance by as much as 66.2% and 58.8%.

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

# A PREDICT + PICK UP OVERVIEW

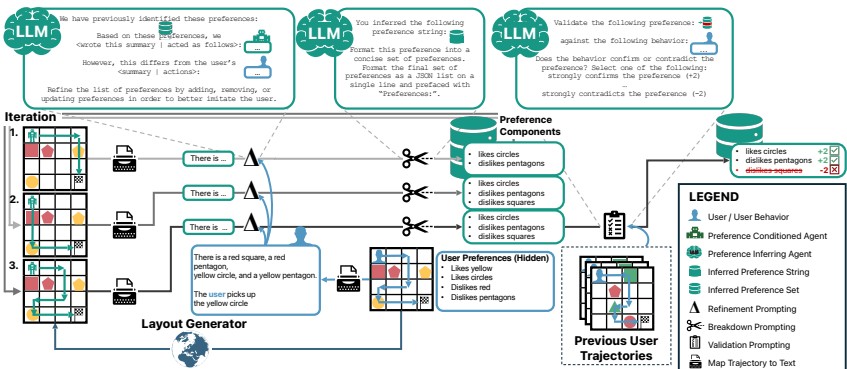

Figure 4: PREDICT Overview. Examples for using PICK UP to infer user preferences are provided for the PLUME. The user has a task they want to provide a demonstration of for PREDICT to learn from. After observing the user's demonstration, PREDICT executes an iterative refinement step (consists of preference update and breakdown) and a validation step. Iterative refinement involves updating the set of inferred references by generating a candidate solution by conditioning the AI assistant on the inferred preference set and prompting an LLM to update the preference set if the candidate solution does not closely match the demonstration. If the preference is updated an LLM is prompted to break it into component parts. Iterative refinement continues until a candidate solution matches the user demonstration. If the preferences were updated in iterative refinement, each preference component is then validated using LLM-as-a-Judge to evaluate how well each component aligns with the user demonstration.

# B METRIC DEFINITIONS

**Preference Inference Quality**

PICKUP:

$$\text{Intersection over Union (IoU)} = \frac{\text{inferred} \cap \text{true}}{\text{inferred} \cup \text{true}}, \tag{1}$$

where inferred is the set of inferred preferences and true is the set of true, target preferences.

PLUME:

$$R_{\text{BERT}} = \frac{1}{|x|} \sum_{x_i \in x} \max_{\hat{x}_j \in \hat{x}} \mathbf{x}_i^\top \hat{\mathbf{x}}_j; \tag{2}$$

$$P_{\text{BERT}} = \frac{1}{|\hat{x}|} \sum_{\hat{x}_j \in \hat{x}} \max_{x_i \in x} \mathbf{x}_i^\top \hat{\mathbf{x}}_j; \tag{3}$$

$$\text{BERTScore (BScore)} = F_{\text{BERT}} = 2 \frac{P_{\text{BERT}} \cdot R_{\text{BERT}}}{P_{\text{BERT}} + R_{\text{BERT}}}, \tag{4}$$

where $x$ is the tokenized reference text (i.e. the true preferences) and $\hat{x}$ is the tokenized candidate text (i.e. the inferred preferences).

**Behavior/Generation Quality**

PICKUP:

$$\text{Return} = \sum_t^{|T|} r(s_t, a_t), \tag{5}$$

where $r(...)$ is the hand coded reward function used to train the human proxy policies, $s_t$ is the state at step $t$, $a_t$ is the action taken at step $t$, and $T$ is a trajectory containing the assistant's solution.

PLUME:

$$\text{PPCM} = \frac{\sum_i^{|\text{true}|} \text{llm\_judge}(\text{true}_i, \text{assistant\_attempt})}{|\text{true}|},\tag{6}$$

where true is the set of true preferences, assistant_attempt is the assistant's summary or email, and llm_judge is a function that prompts the human proxy LLM to evaluate how well a given assistant solution aligns with the true preference on a scale of -2 to +2 (see Appendix Section F.4, Figure 13 for the LLM-as-a-Judge prompt).

# C  ALGORITHM

---

**Algorithm 1** Preference-Conditioned Agent Task Completion

---

1: **Require:**
2: └  $task\_instance \triangleright Task\ instance$
3: Initialize empty preference set $all\_preferences \leftarrow \varnothing$
4: Retrieve relevant examples $related\_examples \leftarrow get\_relevant\_examples(task\_instance.context)$
5: **for** each $example$ in $related\_examples$ **do**
6: └  $all\_preferences \leftarrow all\_preferences \cup example.learned\_preferences$
7: Coalesce and condense preferences $preferences\_to\_use \leftarrow \textbf{\textit{LLM.coalesce}}(all\_preferences)$
8: Generate agent trajectory $agent\_trajectory \leftarrow agent.solve\_task(preferences\_to\_use)$
9: **Output:** Completed task trajectory $agent\_trajectory$ and final preferences $preferences\_to\_use$

---

---

**Algorithm 2** PREDICT: Preference Refinement and Inference

---

1: **Require:**
2: │  $task\_instance \triangleright Task\ instance$
3: │  $agent\_trajectory \triangleright Agent\ trajectory$
4: └  $user\_example \triangleright User\ example$
5: Initialize $inferred\_preference\_set \leftarrow preferences\_to\_use$
6: Set candidate trajectory $candidate\_trajectory \leftarrow agent\_trajectory$
7: **for** each refinement step (up to 3 steps) **do**
8: │  **if** $candidate\_trajectory = user\_example$ **then**
9: │  │  Stop refinement
10: │  **else**
11: │  │  Refine preferences
       $compound\_preference \leftarrow \textbf{\textit{LLM.Refine}}(inferred\_preference\_set,$
                                    $user\_example,$
                                    $candidate\_trajectory)$
12: │  │  Decompose preference
       $inferred\_preference\_set \leftarrow \textbf{\textit{LLM.Breakdown}}(compound\_preference)$
13: │  │  Generate new candidate trajectory
       $candidate\_trajectory \leftarrow agent.solve\_task(inferred\_preference\_set)$
14: Initialize empty validation score list $validation\_scores \leftarrow []$
15: **for** each $preference\_component$ in $inferred\_preference\_set$ **do**
16: │  **for** each $example$ in $related\_examples$ **do**
17: │  │  Validate preference against trajectory
       $new\_score \leftarrow LLM.validate(preference\_component, example)$
18: │  │  $validation\_scores \leftarrow validation\_scores + [new\_score]$
19: │  **if** $mean(validation\_scores) < threshold$ **then**
20: │  │  Discard $preference\_component$
21: Add $task\_instance$ and $inferred\_preference\_set$ to list of examples for future learning

---

## D    PICK UP OBJECTS VISUALIZATION

A rendering of the PICK UP Objects task is provided in Appendix Fig. 5.

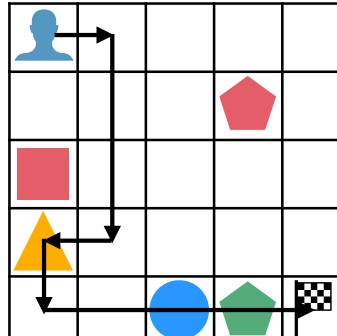

The following objects are available:
a red pentagon, a red square, a yellow triangle,
blue circle, and a green pentagon.

The user picks up the yellow triangle, the blue
circle, and the green pentagon

***Return: 2***

> User Preferences:
>
> "likes yellow", "likes circles", "dislikes red", "dislikes squares"

Figure 5: A depiction of a user example and associated language descriptions for the PICK UP task.

# E EXTENDED RESULTS

Additional results tables and figures discussed in the main body of the paper.

## E.1 PREDICT ITERATIVE STEPS SWEEP

The impact of the number of iterative steps on PREDICT's performance on the two environments and three tasks.

| Iterative Steps | PICK UP | | Summarization | | Emails | |
|---|---|---|---|---|---|---|
| | Jaccard | Return | BScore | PPCM | BScore | PPCM |
| 1 | $0.41_{\pm 0.04}$ | $1.16_{\pm 0.15}$ | $0.26_{\pm 0.01}$ | $0.54_{\pm 0.06}$ | $0.24_{\pm 0.02}$ | $1.21_{\pm 0.13}$ |
| 2 | $0.50_{\pm 0.03}$ | $1.42_{\pm 0.17}$ | $0.26_{\pm 0.02}$ | $0.92_{\pm 0.27}$ | $0.23_{\pm 0.01}$ | $1.18_{\pm 0.17}$ |
| 3 | $0.48_{\pm 0.04}$ | $1.40_{\pm 0.07}$ | $0.23_{\pm 0.01}$ | $0.60_{\pm 0.31}$ | $0.21_{\pm 0.01}$ | $1.32_{\pm 0.22}$ |
| 4 | $0.49_{\pm 0.06}$ | $1.50_{\pm 0.25}$ | $0.23_{\pm 0.03}$ | $1.00_{\pm 0.05}$ | $0.22_{\pm 0.01}$ | $1.30_{\pm 0.17}$ |
| 5 | $0.53_{\pm 0.06}$ | $1.52_{\pm 0.16}$ | $0.22_{\pm 0.03}$ | $0.84_{\pm 0.24}$ | $0.22_{\pm 0.02}$ | $1.10_{\pm 0.36}$ |

Table 2: Iterative Step Sweep. The impact the number of iterative steps has on PREDICT's ability to infer the correct preference set and the quality of generated behaviors across the two environments and three tasks. Results are reported as the mean and standard deviation across three seeds for the following metrics: Jaccard = Jaccard similarity between inferred and true preference sets; BScore=BERTScore.

## E.2 PRELUDE RESULTS

Results on PRELUDE (Gao et al., 2024) for PREDICT and baselines: a No-Learning baseline (NPC), an Oracle preference baseline, in-context learning (ICL), CIPHER-1, and CIPHER-5 Gao et al. (2024) (Table 3). To directly evaluate the ability to infer preferences, we provide all models with ground-truth knowledge of the source of the documents. On the summarization task, PREDICT outperforms all baselines on action/generation quality. On the email writing task, PREDICT outperforms all baselines on the PPCM metric, but slightly underperforms CIPHER-1 on the poorly correlated Levenshtein distance metric (see Section 4.3-**Metric Correlation** for issues with Levenshtein distance).

Results in this table further support issues with the current preference-quality metrics. In the email writing task, the no-learning baseline (which always uses an empty preference), has a higher accuracy than any learning method, which may be due to the significant overlap between preference sets in the task. Further, in both tasks, the highest preference-quality scores do not lead to the highest action-quality scores. We encourage future work to look into alternative preference-quality metrics.

We lastly note that PRELUDE has substantially smaller range between the no-learning (NPC) and oracle preference baselines relative to PLUME. On PPCM, PRELUDE has a range 2.45 and 0.62 for summarization and email writing respectively, while PLUME has ranges of 3.17 and 2.91 for the two tasks. This further supports PLUME as the primary evaluation environment.

| Summarization | | | | |
|---|---|---|---|---|
| Method | Accuracy | BScore | Levenshtein | PPCM |
| No Learning Baselines | | | | |
| NPC | $0.20_{\pm 0.00}$ | $-0.43_{\pm 0.00}$ | $111.50_{\pm 4.91}$ | $-0.89_{\pm 0.12}$ |
| Oracle | $1.00_{\pm 0.00}$ | $1.00_{\pm 0.00}$ | $1.35_{\pm 3.02}$ | $1.56_{\pm 0.09}$ |
| Learning Baselines | | | | |
| ICL | - | - | $113.98_{\pm 7.84}$ | $-0.81_{\pm 0.19}$ |
| C1 | $\mathbf{0.79_{\pm 0.07}}$ | $\mathbf{0.13_{\pm 0.04}}$ | $49.20_{\pm 9.15}$ | $0.79_{\pm 0.24}$ |
| C5 | $0.66_{\pm 0.19}$ | $0.04_{\pm 0.04}$ | $45.34_{\pm 20.07}$ | $0.59_{\pm 0.45}$ |
| PREDICT | $0.73_{\pm 0.10}$ | $0.10_{\pm 0.03}$ | $\mathbf{9.21_{\pm 5.20}}$ | $\mathbf{1.03_{\pm 0.26}}$ |

| Emails | | | | |
|---|---|---|---|---|
| Method | Accuracy | BScore | Levenshtein | PPCM |
| No Learning Baselines | | | | |
| NPC | $0.25_{\pm 0.00}$ | $-0.40_{\pm 0.00}$ | $30.50_{\pm 10.23}$ | $0.89_{\pm 0.09}$ |
| Oracle | $1.00_{\pm 0.00}$ | $1.00_{\pm 0.00}$ | $1.72_{\pm 1.21}$ | $1.51_{\pm 0.03}$ |
| Learning Baselines | | | | |
| ICL | - | - | $35.85_{\pm 9.85}$ | $0.87_{\pm 0.05}$ |
| C1 | $0.07_{\pm 0.07}$ | $-0.25_{\pm 0.03}$ | $\mathbf{13.61_{\pm 6.88}}$ | $0.94_{\pm 0.06}$ |
| C5 | $0.07_{\pm 0.10}$ | $\mathbf{-0.04_{\pm 0.02}}$ | $21.88_{\pm 4.35}$ | $0.93_{\pm 0.12}$ |
| PREDICT | $0.04_{\pm 0.06}$ | $-0.25_{\pm 0.04}$ | $18.93_{\pm 9.10}$ | $\mathbf{0.96_{\pm 0.12}}$ |

Table 3: PRELUDE Results. PREDICT's ability to infer the correct preference set and quality of generated behaviors across the two PRELUDE tasks compared against a no-learning baseline (NPC), a method with access to the true preferences (Oracle), in-context learning (ICL), and CIPHER Gao et al. (2024). Results are reported as the mean and standard deviation across five seeds. Accuracy and Bscore (BERTScore) Zhang* et al. (2020) are preference-quality metrics, while Levenshtein distance and PPCM (per preference-component match) are action-quality metrics.

### E.3   METRIC CORRELATION RESULTS

The metric correlation results for the assistive writing tasks both across the summary versus email writing sub-tasks and by sub-task (Table 4).

| Metric | PRELUDE | | PRELUDE$_{NoEdit}$ | | PLUME | |
|---|---|---|---|---|---|---|
| | Acc. | B.Score | Acc. | B.Score | Acc. | B.Score |
| Emails | | | | | | |
| L-dist | 0.05 | -0.25 | -0.06 | -0.28 | -0.07 | -0.14 |
| ln-L-dist | 0.05 | -0.24 | -0.01 | -0.28 | -0.05 | -0.21 |
| PPCM | 0.29 | **0.34** | 0.21 | **0.30** | 0.42 | **0.77** |
| Summarization | | | | | | |
| L-dist | -0.40 | -0.54 | -0.03 | -0.17 | -0.10 | -0.12 |
| ln-L-dist | -0.50 | -0.60 | -0.18 | -0.37 | -0.16 | -0.37 |
| PPCM | 0.51 | **0.70** | 0.51 | **0.71** | 0.47 | **0.76** |
| Across Tasks | | | | | | |
| L-dist | -0.35 | -0.44 | -0.03 | -0.17 | -0.07 | -0.12 |
| ln-L-dist | -0.43 | -0.48 | -0.11 | -0.22 | -0.15 | -0.30 |
| PPCM | 0.46 | **0.58** | 0.44 | **0.56** | 0.45 | **0.76** |

Table 4: Pearson R correlation between preference similarity metrics and generated writing similarity metrics broken down by task (summarization vs. email). For Levenshtein distance (L-dist) and length-normalized Levenshtein distance (ln-L-dist) lower is better, so inverse correlation is expected. All other metrics are higher is better. Best correlation in each environment is bold. Best overall correlation is underlined. See Section 4.3 for a full description of each metric.

## E.4 PREFERENCE INFERENCE AND CONDITIONING PERFORMANCE BY NUMBER OF USER SAMPLES

In Fig. 6 we show the impact of the number of samples for a given user according to the measures for inferred-preference and action/generation quality metrics.

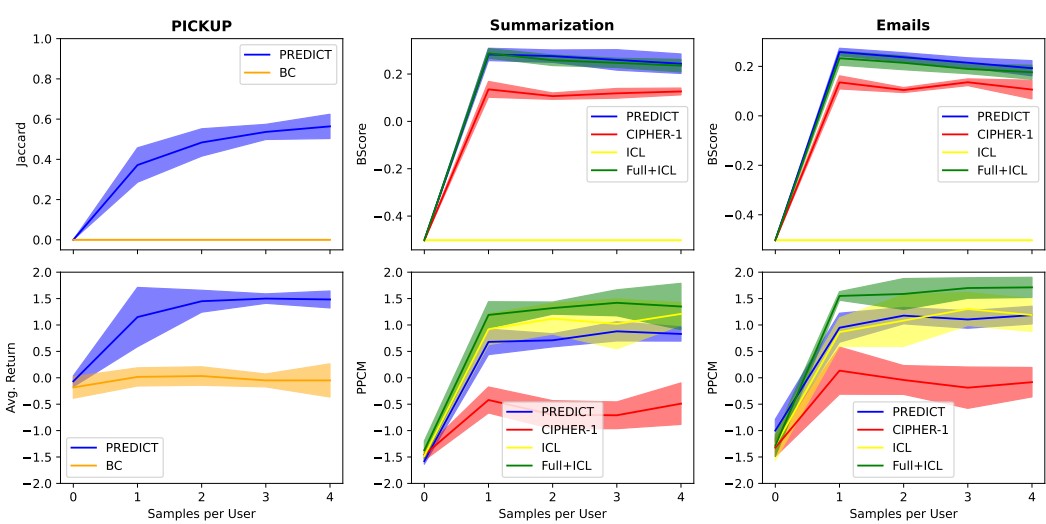

Figure 6: Performance for PREDICT, behavior cloning (BC), CIPHER-1, and in-context learning (ICL) given different numbers of user samples to learn from. Mean and standard deviation (5 seeds) for preference similarity (IoU in PICK UP and BScore in PLUME) and preference-conditioned generation quality (Avg. Return for PICK UP and PPCM for PLUME). GPT-4o is the LLM used.

## F  PRELUDE vs. PLUME PREFERENCE SETS

The preference sets used for each document source and environment (PRELUDE vs. PLUME) are given in Appendix Table 5.

| Document Source | Task Version | User Preferences |
|---|---|---|
| Summarization | | |
| News Articles | PRELUDE | interactive, playful language, positive, short sentences, storytelling, style targeted to young children |
| | PLUME | adopt a step-by-step structure, include a simile, use ampersands (&) instead of "and"s, write in the style of a children's book |
| Chat Forum Posts | PRELUDE | brief, immersive, invoke personal reflection, second person narrative, show emotions |
| | PLUME | adopt a header and sub-header structure, include rhetorical questions, use ALLCAPS to emphasize words, write in the style of a tweet |
| Encyclopedia Pages | PRELUDE | brief, bullet points, parallel structure |
| | PLUME | adopt a rhyming structure, include modern slang, use semicolons (;) when possible, write in the style of a screenplay |
| Paper Abstract | PRELUDE | inquisitive, simple English, skillful foreshadowing, tweet style, with emojis |
| | PLUME | adopt a question-answering style structure, include personifications, use archaic language, write in the style of a podcast |
| Movie Review | PRELUDE | question answering style |
| | PLUME | adopt a stream-of-consciousness structure, include onomatopoeias, use imagery, write in the style of old timey radio |
| Email Writing | | |
| Personal Problem | PRELUDE | conversational, informal, no closing |
| | PLUME | be intensely emotional, include alliterations, use a formal tone, write in a second person narrative |
| Paper Review | PRELUDE | call to action, casual tone, clear, positive |
| | PLUME | be sharply critical, include several short and punchy sentences, use parenthetical asides, write using assertive expressions |
| Paper Tweet | PRELUDE | engaging, personalized, professional tone, thankful closing |
| | PLUME | be blatantly sarcastic, include hyperboles, use an informal tone, write in a third person perspective |
| Paper Summary | PRELUDE | professional greeting and closing, respectful, straight to the points, structured |
| | PLUME | be highly inquisitive, include several long and flowing sentences, use emojis, write using conditional expressions |

Table 5: The user preferences for each assistive writing task (summarization vs. email writing), document source, and version (PRELUDE vs. PLUME).

# G ILLUSTRATIVE EXAMPLES OF ISSUES WITH PRELUDE

## G.1 LEVENSHTEIN DISTANCE AND MULTIPLE GENERATIONS

We present here an inherent issue with Levenshtein distance. For all three generations, we prompt gpt-4o to summarize the encyclopedia article on hotdogs. The first two generations (A. and B.) use identical prompts, in which we ask it to use the preference: "use a haiku". In the third generation, we ask it to use the preference "use a single short sentence". We then calculate the Levenshtein distance between each pair, reported in Table 6. Notably, the Levenshtein distance and length normalized Levenshtein distance, in which a lower score represents higher similarity, would both indicate that generation A and C are more similar to generation A and B, even though generation A and B share the same exact same prompt while generation C uses a entirely different preference. BERTScore does a better job of identifying the most similar texts, however the PPCM scores, which compares writing directly to the true preferences, clearly labels both Generation A. and Generation B. as "strongly exhibiting" the preference "use a haiku", and Generation C. as "strongly contradicting" the preference.

Generation A.

> A sausage in bun,
> Toppings vary near and far,
> Hot dog, rich in roots.

Generation B.

> Grilled or steamed sausage,
> In a bun with cheese or spice,
> Street food's warm embrace.

Generation C.

> A hot dog is a sausage in a bun with various toppings such as mustard, ketchup,
> and relish

| Pairing | Levenshtein Dist. | Norm. Levenshtein Dist. | BERTScore | PPCM per generation |
|---------|-------------------|-------------------------|-----------|---------------------|
| A-B     | 17                | 0.85                    | 0.27      | (2, 2)              |
| A-C     | 16                | 0.76                    | 0.20      | (2, -2)             |
| B-C     | 17                | 0.81                    | 0.10      | (2, -2)             |

Table 6: Levenshtein distance between each pair of generations.

## G.2 EDITING INFLUENCES THE USER

In this next example, we demonstrate how the editing procedure can influence the output of the user. We again ask an LLM to summarize the encyclopedia article on hot dog. Let's assume, an inferring agent has incorrectly inferred the preference "question answering style". Using this preference, the preference-conditioned agent writes:

> What is a hot dog?
>
> A hot dog is a dish made of a sausage (typically a wiener or frankfurter) served in a partially sliced bun. It can refer to the sausage itself, which is grilled, steamed, or boiled.
>
> How is a hot dog typically prepared?
>
> Hot dog preparation varies, but common condiments include mustard, ketchup, relish, and onions. Some toppings can also include cheese sauce, sauerkraut, chili, bacon, or jalapeños.
>
> Where did hot dogs originate?
>
> The sausages used in hot dogs were culturally imported from Germany, and the dish became popular in the U.S. as a working-class street food. Over time, hot dogs became closely associated with American culture, particularly baseball.
>
> Where are hot dogs most popular?
>
> While they are especially associated with New York City, hot dogs became widespread across the United States throughout the 20th century, becoming an integral part of American cuisine.

We ask the user if the above is satisfactory for it's true preference of: "style targeted to young children". It does not, so we then ask the user to edit the above for it true preference "style targeted to young children", and the user writes:

> What is a hot dog?
>
> A hot dog is a yummy snack made of a sausage (like a wiener or frankfurter) inside a soft bun. Sometimes people call the sausage itself a hot dog, and it can be cooked by grilling, steaming, or boiling.
>
> How is a hot dog usually made?
>
> Hot dogs can be made in different ways, but they often have yummy toppings like mustard, ketchup, relish, and onions. Some people like to add even more fun stuff like cheese, sauerkraut, chili, bacon, or spicy jalapeños.
>
> Where did hot dogs come from?
>
> The sausages in hot dogs came from Germany, and a long time ago, they became super popular in the U.S. as a quick and tasty street food for busy people. Now, hot dogs are a big part of American culture, especially at baseball games!
>
> Where are hot dogs the most popular?
>
> Even though hot dogs are popular all over, they are especially loved in New York. Over the years, hot dogs have become a favorite food all across the United States!

While the above summary does correctly follow the true preference, it is still formatted in a question answering style, which is not part of the user's preferences. I.e., the preference-inferring agent's error has directly impacted the user's writing. This is another reason why PLUME removes the editing procedure.

For reference, if the user is directly asked to summmarize the encyclopedia article on hot dogs using it's true preference of "style targeted to young children", it outputs:

> A hot dog is a yummy food that's made with a sausage, like a wiener or frankfurter, inside a soft bun. You can cook the sausage in different ways, like grilling or boiling. People love to put tasty toppings on their hot dogs, such as ketchup, mustard, cheese, and even onions!
>
> Hot dogs came from Germany, but they became super popular in the US, especially at baseball games. Now, you can find hot dogs all over the country, and in some places, like Chicago, people have special ways of making them!

# H PROMPTS

## H.1 PREFERENCE INFERENCE AND PREFERENCE-CONDITIONED AGENT PROMPTS

The prompts used by PREDICT for candidate trajectory generation and task completion in the PLUME environment are in Appendix Fig. 7. The prompts used by PREDICT to infer the user preferences from user examples are provided in Appendix Fig. 9 for the PICK UP environment and in Appendix Fig. 8 for the PLUME environment.

---

**System Prompt**

You are an experienced writer. Adapt your writing to heavily emphasize the provided preferences.

**User Prompt**

You have the following preferences: [`<inferred_preference_1>`,..., `<inferred_preference_k>`]

Using these preferences, write a short {`summary` | `email`} about {`this` | `these`} {`article` | `notes`}:

[START OF {ARTICLE | NOTES}]
`<task_content>`
[END OF {ARTICLE | NOTES}]

Encapsulate the {`summary` | `email`} in triple quotes
"""
<{`summary` | `email`}>
"""

---

Figure 7: LLM prompts for the **preference-conditioned agent** and for **task completion** on the **PLUME's** summarization and e-mail writing tasks. The system prompt is prepended to the user prompt following the LLM's chat template. "{...|...}" means that of the two options is selected based on the task and "<...>" indicates that the text is formatted from a variable. `inferred_preference_i` refers to one of the inferred user preferences.

**System Prompt**

A user is completing writing tasks. The user has an underlying set of preferences that explains why they write the way they do.

**User Prompt**

**Aggregation Task**

We are tasked to curate a prompt to guide a specific style of writing. We currently have the following list of preferences related to writing styles:
[<inferred_preference_1>,..., <inferred_preference_l>]
Unfortunately, these preferences may overlap or contain redundancies. Please review the list and condense it by combining similar or overlapping preferences, ensuring that the distinct intent behind each one remains clear so that a writer can easily follow them. Ensure the condensed list is concise, non-redundant, and preserves the original level of specificity. When applicable, preserve the exact wording. Return the revised preferences in the same format as the original list.

**Inference Task**

We received a new task. The task is to {summarize | write an email about} the following:
<article | notes>

We have previously identified the following preferences:
[<inferred_preference_1>,..., <inferred_preference_k>]
Based on these preferences, we wrote this {summary | email}:
<assistant_output>

However, this differs from the user's {summary | email}. The user wrote this {summary | email}:
<user_output>

Refine the list of preferences by adding, removing, or updating preferences in order to better imitate the user.
While refining the preference set, you should:
- Identify and reason about differences between our writing and the user's writing.
- Consider writing traits from distinct quirks to broader stylistic tendencies.
- Provide a concise set of preferences in the imperative form.
- Be precise; make the fewest possible changes to the preference set.
- Do not qualify, dilute, or soften existing preferences.
- Only refine the preferences if a clear difference exists. Otherwise, preserve the current preferences.

Provide a concise set of specific preferences in the imperative form. After reasoning, output the refined set of preferences as a JSON array, where each element is a string, on a single new line and prefaced with "Preferences:".

Figure 8: LLM prompts for **preference inference** on **PLUME's** summarization and e-mail writing tasks. The system prompt is prepended to each user prompt following the LLM's chat template. "{...|...}" means that of the two options is selected based on the task and "<...>" indicates that the text is formatted from a variable. user_output refers to how the user completes the task, assistant_output how the assistant completes the task, and inferred_preference_i to one of the inferred user preferences. Continued on next page.

---

**User Prompt**

**Preference Breakdown Task**

You inferred the following preference string:
`[<inferred_preference_1>,..., <inferred_preference_k>]`
Format this preference into a concise set of preferences. Format the final set of preferences as a JSON list on a single line and prefaced with "Preferences:". Each element in the JSON list should be a string.The final output should look like: Preferences: [<preference 1>,..., <preference i>, ...]

**Validation Task**

Validate the following preference: "[`<inferred_preference_1>`, ..., `<inferred_preference_k>`]" against the following writing:

`<user_output>`

Does the writing confirm or contradict the preference? Select one of the following: strongly confirms the preference, somewhat confirms the preference, is neutral toward the preference, somewhat contradicts the preference, strongly contradicts the preference. Your final decision should be output on a separate line prefaced with "Verdict:".

Figure 8: LLM prompts for **preference inference** on the **PLUME's** summarization and e-mail writing tasks. The system prompt is prepended to each user prompt following the LLM's chat template. "{...|...}" means that of the two options is selected based on the task and "<...>" indicates that the text is formatted from a variable. `user_output` refers to how the user completes the task, `assistant_output` how the assistant completes the task, and `inferred_preference_i` to one of the inferred user preferences.

---

---

**System Prompt**

A user is completing tasks where they pick up objects of different colors and shapes. The user has an underlying set of preferences that explains why they pick up the objects they do. The objective is to identify these underlying preferences so that we can act exactly like the user.

**User Prompt**

**Inference Task**

We received a new task.

In this task, the following objects are available: a green square, a red pentagon, a red square, a yellow circle, and a yellow square.

We have previously identified the following preferences: [<inferred_preference_1>,...,<inferred_preference_k>] Based on these preferences, <agent_output>.

However, this differs from the user's actions. <user_output>.

Refine the list of preferences by adding, removing, or updating preferences in order to better imitate the user.

While refining the preference set, you should:
- Reason about the difference between the objects we selected and the objects the user selected.
- Make the fewest changes to the preference set to improve our actions.
- Consider both the objects that were selected and those that were not selected.
- Reason about the specific shapes that the user may like or dislike and the specific colors that the user may like or dislike.
- Think step by step.

After reasoning, output the refined preference on a new line and prefaced with "Preferences:".

**Preference Breakdown Task**

You inferred the following preference string:
"[<inferred_preference_1>,..., <inferred_preference_k>]"
Format this preference into a concise set of preferences.

Format the final set of preferences as a JSON list on a single line and prefaced with "Preferences:". Each element in the JSON list should be a string with exactly two words in the format "<likes/dislikes> <attribute>" where <attribute> must be a single shape or color. Putting this together, the final output should look like:

Preferences: ["likes <color/shape>", ..., "dislikes <color/shape>", ...]

---

Figure 9: LLM prompts for each step of **preference inference** on the **PICK UP** task. The system prompt is prepended to each user prompt following the LLM's chat template. "<...>" indicates that the text is formatted from a variable. user_output refers to how the user completes the task, assistant_output how the assistant completes the task, and inferred_preference_i to one of the inferred user preferences. Continued on next page.

> **User Prompt**
>
> > **Validation Task**
> >
> > Validate the following preference: "[`<inferred_preference_1>`,...,
> > `<inferred_preference_k>`]" against the following behavior:
> >
> > `<state_definition>` `<user_output>`
> >
> > Does the behavior confirm or contradict the preference?
> > Select one of the following: strongly confirms the preference, somewhat confirms the preference, is neutral toward the preference, somewhat contradicts the preference, strongly contradicts the preference.
> >
> > While validating the preference, you should:
> > - Think step by step.
> >
> > The final verdict should be output on a separate line in the format:
> > Verdict: confirms/contradicts/neutral

Figure 9: LLM prompts for each step of **preference inference** on the **PICK UP** task. The system prompt is prepended to each user prompt following the LLM's chat template. "`<...>`" indicates that the text is formatted from a variable. `user_output` refers to how the user completes the task, `assistant_output` how the assistant completes the task, and `inferred_preference_i` to one of the inferred user preferences.

## H.2 SYNTHETIC HUMAN PROMPTS

The prompts used to have GPT-4o play the role of our synthetic human for PREDICT are given in Appendix Fig. 10. The "human" is instructed to complete the task in the same way as the preference-conditioned agent when completing the writing tasks (see Appendix Fig. 7).

---

**System Prompt**

You are an experienced writer. Adapt your writing to heavily emphasize the provided preferences.

**User Prompt**

You have the following preferences: [<inferred_preference_1>,..., <inferred_preference_k>]

Using these preferences, write a short {summary | email} about {this | these} {article | notes}:

[START OF {ARTICLE | NOTES}]
<task_content>
[END OF {ARTICLE | NOTES}]

Encapsulate the {summary | email} in triple quotes
"""
<{summary | email}>
"""

---

Figure 10: LLM prompts for the **synthetic human** on the **PLUME's** summarization and e-mail writing tasks. The system prompt is prepended to the user prompt following the LLM's chat template. "{...|...}" means that of the two options is selected based on the task and "<...>" indicates that the text is formatted from a variable. inferred_preference_i refers to one of the inferred user preferences.

### H.3 PREFERENCE-CONDITIONED AGENT BASELINE PROMPTS

The prompts used in the no-preference baseline are in Appendix Fig. 11 and for the in-context learning baseline are in Appendix Fig. 12. For the in-context learning baseline, the number of examples $l$ matches the number of examples used when coalescing prevoiusly inferred prompts (see Appendix Fig. 8).

---

**System Prompt**

You are an experienced writer. Adapt your writing to heavily emphasize the provided preferences.

**User Prompt**

Write a short {summary | email} about {this | these} {article | notes}:

[START OF {ARTICLE | NOTES}]
<task_content>
[END OF {ARTICLE | NOTES}]

---

Figure 11: LLM prompts for the **no preference baseline** in the **PLUME environment**. The system prompt is prepended to the user prompt following the LLM's chat template. "<...>" indicates that the text is formatted from a variable. task_content refers to the content of either the article to be summarized or the notes to include in the email, depending on the sub-task.

> **System Prompt**
>
> You are an experienced writer. Adapt your writing to heavily emphasize the provided preferences.

> **User Prompt**
>
> You have previously observed the following examples:
>
> Example 0:
> {Article | Notes}:
> [START OF {ARTICLE | NOTES}]
> `<task_content>`
> [END OF {ARTICLE | NOTES}]
>
> {Article | Notes}:
> """"""
> `<completion_0>`
> """"""
>
>
> .
> .
> .
>
>
> Example $l$:
> {Article | Notes}:
> [START OF {ARTICLE | NOTES}]
> `<task_content>`
> [END OF {ARTICLE | NOTES}]
>
> {Article | Notes}:
> """"""
> `<completion_l>`
> """"""
>
> Using the same style as these examples, write a short {`summary` | `email`} about {`this` | `these`} {`article` | `notes`}:
>
> [START OF {ARTICLE | NOTES}]
> `<task_content>`
> [END OF {ARTICLE | NOTES}]
>
> Encapsulate the {`summary` | `email`} in triple quotes
> """"""
> `<{summary | email}>`
> """"""

Figure 12: LLM prompts for the **in-context learning baseline** in the **PLUME environment**. The system prompt is prepended to the user prompt following the LLM's chat template. "`<...>`" indicates that the text is formatted from a variable, and `completion_l` refers to an example completion provided for in-context learning. `task_content` refers to the content of either the article to be summarized or the notes to include in the email, depending on the sub-task.

## H.4 LLM-AS-A-JUDGE PROMPTS

The prompts used by the LLM-as-a-Judge are shown in Fig. 13.

---

**System Prompt**

You are an experienced editor that is evaluating writing samples.

**User Prompt**

You received the following {`summary`|`email`}:
"""
`<agent_completion>`
"""
Does the above {`summary` | `email`} exhibit the following preference: `<true_preference_i>`?
Identify, analyze, and reason about specific excerpts that show similarities or contradictions of underlying preferences. After reasoning, select one of the following options:
clearly exhibits, somewhat exhibits, neither exhibits nor contradicts, somewhat contradicts, clearly contradicts
Your final selection should be on a new line prefaced with "Verdict:"

---

Figure 13: **LLM-as-a-Judge prompts** for the per preference-component match metric (PPCM) used in the **PLUME environment**. The system prompt is prepended to the user prompt following the LLM's chat template. "`<...>`" indicates that the text is formatted from a variable. `agent_completion` refers to the agent's article summary or email, depending on the sub-task. `true_preference_i` refers to one of the $k$ true preferences that the user has.

