# OpenReview forum: "PREDICT: Preference Reasoning by Evaluating Decomposed preferences Inferred from Candidate Trajectories"
_ICLR.cc/2025/Conference — Submitted to ICLR 2025_

### Official Review · Reviewer_Cces · 2024-11-01

**Soundness:** 3
**Presentation:** 2
**Contribution:** 3
**Rating:** 6
**Confidence:** 3

**Summary:**

This paper proposes a prompting-based method for inferring and refining user preferences from action trajectories. The proposed method takes an initial set of inferred preferences, asks a language model to refine them (e.g., by collapsing duplicates, breaking down complex preferences into individual preferences) and then validates the preferences by asking a language model whether sampled user examples confirm or contradict the inferred preference. The paper tests the proposed approach in two domains, (1) a gridworld object collection domain and (2) a writing/editing benchmark, finding clear gains, especially in the latter domain.

**Strengths:**

1. The proposed method is relatively simple but appears to provide clear gains on the preference inference task
2. The paper provides two new tasks, one toy gridworld (PICK UP) and one modification of an existing assistive writing task (PLUME). These tasks, especially PLUME, seem like good starting points for future modeling work

**Weaknesses:**

1. The proposed method and benchmark could be a bit more clearly described. In particular, I found the interleaving of contributions and the actual method description in Section 3 to be a bit confusing, and it’s a bit difficult to understand how the iterative refinement step works without looking at the corresponding prompts in the Appendix — perhaps a more detailed Figure 2 could help here? I also think the description of PREDICT is a bit hard to follow for someone not familiar with PLUME (and again, I think figures would help!)
2. Not really a weakness (i.e., does not affect my score): the paper only focuses on inferred preferences, rather than stated preferences. It would be great to see how this approach works in a setting with both stated and inferred preferences

**Questions:**

1. One concern I had with the preference validation approach is that preferences might only apply in rare circumstances. It seems that the validation stage would then return almost all neutrals, which would result in filtering out, even if the preference was very clear. Is this reasoning accurate and is there a way to address this issue (e.g., by doing some more targeted sampling)?
2. Honestly, I am not sure why performance on the PICK UP task increases, given the simplicity of preferences in that environment. One potential concern would be that if initial inferred preferences are over objects (e.g., “prefers yellow triangles”), then the breaking down of preferences could provide an implicit bias toward the “compositional” preference structure of the task. Can you provide a bit more intuition on whether this is a valid concern, and why you think performance on the PICK UP task improves?
3. Would it be possible to provide some examples of inferred preferences before and after applying your method somewhere in the paper?

---

> ### Author Response · Authors · 2024-11-15
> **Author Response**
>
> **Weaknesses**
> > The proposed method and benchmark could be a bit more clearly described. In particular, I found the interleaving of contributions and the actual method description in Section 3 to be a bit confusing, and it’s a bit difficult to understand how the iterative refinement step works without looking at the corresponding prompts in the Appendix — perhaps a more detailed Figure 2 could help here? I also think the description of PREDICT is a bit hard to follow for someone not familiar with PLUME (and again, I think figures would help!)
>
> Thank you for pointing this out. Other reviewers raised similar concerns and with hindsight we agree it is currently difficult to concretely understand the algorithm. To improve the description of the method, we updated Figure 2 (now Figure 1 in the paper update via rebuttal revision) to contain more details and rewritten the **Iterative Refinement** section (now pages 3-4 in the paper update via rebuttal revision). We would appreciate any further suggestion on how to improve PREDICT's description.
>
> > Not really a weakness (i.e., does not affect my score): the paper only focuses on inferred preferences, rather than stated preferences. It would be great to see how this approach works in a setting with both stated and inferred preferences
>
> Thank you for calling out the role for stated preferences. This is future work we are excited to address. A simple way to account for stated preferences in PREDICT would be to collect them from the human, assume they are accurate, and then directly and permanently add them to the preference set. The stated preferences would then be used whenever the assistant was solving a task for the human and during the counterfactual step of preference inference. We did not run experiments with this set up as it would move performance closer to the oracle baseline, only because we would have access to a subset of the ground truth preferences. Future work would be to identify how to resolve potential errors or misspecifications in the stated preferences.
>
> **Questions**
>
> > One concern I had with the preference validation approach is that preferences might only apply in rare circumstances. It seems that the validation stage would then return almost all neutrals, which would result in filtering out, even if the preference was very clear. Is this reasoning accurate and is there a way to address this issue (e.g., by doing some more targeted sampling)?
>
> For PREDICT, whether a rare preference is retained depends on the examples retrieved during validation (lines 190 - 191 in the attached and updated draft and Section 4.3 “Knowledge of Contexts”). So long as the examples relevant to the rare preference are retrieved, the rare preference will be retained. This would rely on the “more targeted sampling” approach you reference in your question.
> In this paper the issue of high quality retrieval is out of scope in order to focus on the quality of the preference inference mechanism, so we use an oracle context retrieval method. However, it is an important avenue for future work and rare preferences is an important factor to design and test for.
>
> > Honestly, I am not sure why performance on the PICK UP task increases, given the simplicity of preferences in that environment. One potential concern would be that if initial inferred preferences are over objects (e.g., “prefers yellow triangles”), then the breaking down of preferences could provide an implicit bias toward the “compositional” preference structure of the task. Can you provide a bit more intuition on whether this is a valid concern, and why you think performance on the PICK UP task improves?
>
> We agree the compositional nature of the PICKUP task does lead to an implicit bias towards compositional preferences. The compositional nature of the PICKUP tasks only impacts our ablation over PREDICT’s different components, and not any of the baselines. Our PREDICT_{CP} ablation, looking at the impact of breaking preferences down, was to verify the step was helping in a scenario where we know it should. We can make this more clear in our discussion of the results.
>
> We believe the improvement in performance relative to baselines is due to the partial observability of the task, which implies there are multiple potential explanations and solutions. The counterfactual and iterative refinement allows the system to attempt a few different potential solutions and test its hypotheses against the environment to see which best approximates the user’s intentions. The baselines do not have this ability to test hypotheses. The partial observability can lead to inferring ambiguous preferences, and the validation step makes it possible to reduce how ambiguous an inferred preference is.

---

> > ### Comment · Reviewer_Cces · 2024-11-25
> >
> > > we updated Figure 2
> >
> > I agree with Reviewer tU2z that this version of the figure is actually more confusing -- it just has so much info, it takes a really long time to read and understand what's going on. I'd consider at minimum removing one of the two tasks (e.g. PICK UP) from the figure, but maybe simplifying it even further is necessary. Also, if you want, you could put a PICK UP version of the figure in the appendix. The writing in the corresponding section is definitely better, now, though!
> >
> > Thank you for answering my questions too! I'll keep my score the same for now, but still recommend acceptance if the other reviewers are in agreement.

---

> > > ### Author Response · Authors · 2024-11-25
> > >
> > > Thank you for feedback on our main figure. Based on your feedback and that of reviewer tU2z, we have simplified the figure. Now an example is provided only for the writing task and the example for PICK UP is moved to Appendix A.
> > >
> > > Please let us know if this updated version does not address your concerns.

---

> > > > ### Comment · Reviewer_Cces · 2024-11-27
> > > >
> > > > I do think the figure is still a bit too busy, and could be further improved. Here are some suggestions:
> > > >
> > > >  - Right now, there are too many arrows going all over the place, and it's really hard to understand the high-level information flow of the figure. Having more clear demarcations of inputs / outputs would be useful
> > > > - In general, I think there's just too much going on in this figure, and having the separation between general prompts and the specific examples probably adds more complexity than it is worth
> > > > - Ideally, I feel like the figure would just have 3-4 boxes with arrows between them
> > > > - It's also possible that you could abstract away some of the details, e.g., breaking a preference description into component parts doesn't necessarily have to be included imo, and you could just describe it in the figure caption/body of the paper
> > > > - I would recommend simplifying the figure enough that you don't need a legend at all. In my opinion, the goal here should really be to have a figure that someone who isn't familiar with the paper can skim and understand in just a few seconds!
> > > >
> > > > This doesn't affect my score, fyi, but hopefully some of these tips are helpful?

---

> ### Author Response · Authors · 2024-11-15
> **Author Response 2**
>
> > Would it be possible to provide some examples of inferred preferences before and after applying your method somewhere in the paper?
>
> Yes, here are some examples of the preferences inferred at each step in PREDICT:
>
> **BOLD** indicates a new component that moves the inferred preferences toward the true preferences
>
> *ITALICS* indicates a new component that moves the inferred preferences away from the true  preferences
>
> 🔥 indicates overfitting to a specific example
>
>
> PLUME’s Summarization Task
>
>
> TRUE PREFERENCES (Hidden from the agent):
>
> - include a simile;
> - adopt a step-by-step structure;
> - use ampersands (&) instead of "and"s;
> - write in the style of a children's book
>
> EXAMPLE 1:
>
> step 1
>
> - **"write in a narrative style, like telling a story."**,
> - **"use simple, straightforward language."**,
> - **"break down the summary into clear, numbered steps."**,
> - "maintain an enthusiastic and positive tone.",
> - 🔥"refer to individuals by their first names to personalize the summary."
> - 🔥"include direct quotes from key figures.",
>
> step 2
>
> -"write in a narrative style, like telling a story",
> -"use simple, straightforward language",
> -"break down the summary into clear, **playful steps using 'Step'"**,
> -"maintain an enthusiastic and positive tone",
> -"refer to individuals by their first names after the initial full name mention to personalize the summary",
> -"include direct quotes from key figures",
> -🔥"include specific examples and anecdotes to add color to the story"
>
>
> step 3
>
>   - "write in a narrative style, like telling a story",
>  -  **"use playful and whimsical language"**,
>   - "break down the summary into clear, **concise steps using 'step'"**,
>  -  "maintain an enthusiastic and positive tone",
>   - "refer to individuals by their first names after the initial full name mention"
>   - "include direct quotes from key figures",
>   - "include specific examples and anecdotes to add color to the story",
>
>
>
> EXAMPLE: 2
>
> Starting preferences:
>
> - "write in a narrative style, like telling a story"
> - "use playful and whimsical language"
> - "break down the summary into clear, concise steps using 'step'"
> - "maintain an enthusiastic and positive tone"
> - "refer to individuals by their first names after the initial full name mention"
> - "include direct quotes from key figures”
> - "include specific examples and anecdotes to add color to the story"
> - "use a fairy tale or storybook style"
> - "simplify scientific details in a straightforward manner"
>
>
> step 1
> - *"simplify language and avoid overly whimsical descriptions",*
> - *"maintain a casual and straightforward tone",*
> - **"use a step-by-step structure with concise steps"**,
> - 🔥"use short and direct quotes",
> - "include specific examples and anecdotes",
> - "refer to individuals by their first names after the initial full name mention"
>
> step 2
>
> - "simplify language and **use informal, relatable expressions"**,
> - **"maintain a whimsical and story-like tone"**,
> - "use a step-by-step structure with numbered steps",
> - "use short and direct quotes",
> - "include specific examples and anecdotes",
> - "refer to individuals by their first names after the initial full name mention",
> - **"use symbols like '&' instead of 'and'"**
>
> step 3
> - "simplify language and use informal, relatable expressions",
> - "maintain a whimsical and story-like tone",
> - "use a step-by-step structure with numbered steps",
> - "use short and direct quotes",
> - "include specific examples and anecdotes",
> - "refer to individuals by their first names after the initial full name mention",
> - "use symbols like '&' instead of 'and'",
> - **"use analogies and metaphors"**,
> - 🔥"use exclamation marks for emphasis"

---

> ### Author Response · Authors · 2024-11-21
> **Author Response Follow Up**
>
> Hello,
>
> As we are over halfway through the discussion period, we are reaching out to check in on whether our responses have addressed your concerns. Please let us know if we have not, and what we can do to better address your concerns.
>
> Thank you.

---

### Official Review · Reviewer_cehp · 2024-11-03

**Soundness:** 3
**Presentation:** 3
**Contribution:** 3
**Rating:** 6
**Confidence:** 2

**Summary:**

This paper presents PREDICT (Preference Reasoning by Evaluating Decomposed Preferences Inferred from Candidate Trajectories), a novel method aimed at enhancing the inference of human preferences in AI agents. Addressing the limitations of existing Large Language Models (LLMs) that often generate broad and generic preferences, PREDICT introduces three key innovations: iterative refinement of inferred preferences, decomposition of preferences into constituent components, and validation of preferences across multiple user trajectories. The approach is rigorously evaluated in two distinct environments—a gridworld setting (PICK UP) and a newly introduced text-domain environment (PLUME)—demonstrating substantial improvements over baseline methods by 66.2% in PICK UP and 41.0% in PLUME. Additionally, augmenting PREDICT with in-context learning yields a further 17.9% enhancement, underscoring its effectiveness in capturing nuanced human preferences.

**Strengths:**

1. Good Solution: introduce a thoughtful combination of iterative refinement, preference decomposition, and validation, effectively addressing the generic nature of existing preference inference methods.

2. A new reliable benchmark: LUME environment is a valuable contribution, offering a more reliable benchmark compared to previous setups like PRELUDE.

3. Significant Performance Gains: The method achieves impressive improvements over established baselines, notably outperforming behavioral cloning by 66.2% in PICK UP and CIPHER by 41.0% in PLUME, highlighting the practical efficacy of the proposed components.

**Weaknesses:**

1. Need investigate how varying iterative refinement steps impacts the performance and efficiency of the method, potentially overlooking optimal configurations.
2. Lack of Real User Case: experiments are primarily based on synthetic users, consider adding some real case analyses to verify the practicality and generalizability of the method.

**Questions:**

See weakness part.

---

> ### Author Response · Authors · 2024-11-15
> **Author Response**
>
> **Weaknesses/Questions**
>
> > Need investigate how varying iterative refinement steps impacts the performance and efficiency of the method, potentially overlooking optimal configurations.
>
> We thank the reviewer for raising this point. When setting up our experimental conditions for PREDICT, we were originally testing the framework with Llama3 70B-Instruct and found there were significant improvements during the first three iterations, but much more modest improvements thereafter. In PLUME, we were also occasionally observed overfitting to specific examples with more than 3 iterations. In light of the computational cost of further refinements and the limited improvements observed after the third iteration, we set the number of iterations to 3.
>
> For this rebuttal, we have swept between 1 - 5 refinement steps for both PICKUP and PLUME with GPT4-o as the underlying LLM. The results from the sweep are reported on page 17 of the updated paper (via rebuttal revision). These new results show the highest performance with 4 steps. We are currently re-running our PREDICT ablations with 4 steps and will report the results with an updated Table 1 as soon as we have them.
>
> > Lack of Real User Case: experiments are primarily based on synthetic users, consider adding some real case analyses to verify the practicality and generalizability of the method.
>
> When writing the paper, we considered how to run a human evaluation, but ran into several experiment design challenges that we identified as factors that would lead to noisy results. For example, where the LLM is already biased towards a human user’s preferences all methods perform essentially the same; if the LLM is biased towards concise and grammatical generations then correctly inferring these preferences will not change the LLM’s behavior. In order to see the impact of “learning” we need to ensure that the human users have preferences that are not already baked into the assistant LLM.  It is challenging to recruit human participants that capture the range of possible preferences, especially those not an LLM is not already biased to. We could assign human preferences we have verified that are not baked into the model. However, we are unsure how to guarantee that the human users would be able to produce demonstrations that comply with their assigned preferences. Therefore, we agree that real human experiments on PREDICT would be a useful contribution, but leave resolving the aforementioned issues required for human evaluation for future work.
>
> We will update the discussion with the difficulties of human evaluations due to: PREDICT and methods like being designed to induce the preferred behaviors/generations in an assistant LLM that the assistant LLM is not already biased towards.

---

> ### Author Response · Authors · 2024-11-21
> **Author Response Follow Up**
>
> Hello,
>
> As we are over halfway through the discussion period, we are reaching out to check in on whether our responses have addressed your concerns. Please let us know if we have not, and what we can do to better address your concerns.
>
> Thank you.

---

> ### Author Response · Authors · 2024-11-25
>
> Hello,
> We have less than two days left in the discussion period.
>
> Based on feedback from reviewers Cces and tU2z, we have simplified our main figure. Now an example is provided only for the writing task in the main body of the paper, and the example for PICK UP is moved to Appendix A.
>
> Please let us know if this updated version and our responses above have not addressed your concerns.
>
> Thank you.

---

> ### Author Response · Authors · 2024-11-25
>
> Hello,
> We have less than two days left in the discussion period.
>
> Based on feedback from reviewers Cces and tU2z, we have simplified our main figure. Now an example is provided only for the writing task in the main body of the paper, and the example for PICK UP is moved to Appendix A.
>
> Please let us know if this updated version and our responses above have not addressed your concerns.
>
> Thank you.

---

### Official Review · Reviewer_27u7 · 2024-11-04

**Soundness:** 2
**Presentation:** 2
**Contribution:** 2
**Rating:** 5
**Confidence:** 3

**Summary:**

The paper proposed a method for personalized LLM assistant that infers implicit user preference. The PREDICT pipeline includes an initial LLM assistant which conducts the same task as the user, a preference-inferring agent which summarizes the difference between the user and the assistant's trajectory, as well as a validation step to generalize the inferred preferences on multiple users.

The paper conducted experiments in two environment settings, a grid world navigation, and PLUME, a text summarization and email editing environment. The paper proposed two evaluation dimensions, preference quality and action quality, to measure the inferred preference correctness and the overall task completion rate. Both evaluation dimensions are further translated in domain-specific measures.

The paper compared the proposed method against several baselines and conducted ablation studies, and demonstrated improved performance.

**Strengths:**

- The paper studies an important and challenging problem: how to infer and accommodate user's implicit preferences
- The paper proposed a method to infer user's preference by comparing user VS LLM assistant's action trajectories, decompose into several components, and validate the preferences across users
- The paper conducted experiments in two environments
- The paper conducted thorough experiments with ablation studies and compared against several baselines

**Weaknesses:**

The paper could largely benefit from a major round of revision:
- It would be helpful to illustrate detailed examples for each of the benchmark environment in the main paper
- Figure 2 is very abstract. It would be helpful to add more details, such as examples of the constituent components, examples of the contrast between user and agent's predictions, examples of validation set, etc.
- It would be helpful to formally define each of the evaluation metrics
- Table 1: how significant are there results as most of the differences fall within the std margin. How would Table 1 support the main claim (Ln 22) that 'PREDICT more accurately infers nuanced human preferences', especially when the Ln 420 concluded that 'PPCM does not show a clear trend' when comparing PREDICT to PREDICT_{CP}
- The paper heavily references Gao et al. 2024 for experiment setups, methods, etc without specifying the necessary details within the paper

**Questions:**

1. Ln 142 "base components": are they all hand crafted for each task?
2. Ln 156, 173, 260: who are the 'users'? Human? LLM proxy?
3. Ln 233: it is unclear what the relevance/importance it is for the environment to be partially observable

---

> ### Author Response · Authors · 2024-11-15
> **Author Response**
>
> **Weaknesses**
> > It would be helpful to illustrate detailed examples for each of the benchmark environment in the main paper
>
> We have updated Figure 2  (now Figure 1 in the paper update via rebuttal revision) to include specific examples of PICKUP and PLUME (see attached PDF “PREDICT_detailed_main_figure”). Does the modified figure address your concerns?
>
> > Figure 2 is very abstract. It would be helpful to add more details, such as examples of the constituent components, examples of the contrast between user and agent's predictions, examples of validation set, etc.
>
> Thank you for pointing this out. Other reviewers raised similar concerns and with hindsight we agree it is currently difficult to concretely understand the algorithm. To improve the description of the method, we updated Figure 2 (now Figure 1 in the paper update via rebuttal revision) to contain more details and rewritten the Iterative Refinement section (now pages 3-4 in the paper update via rebuttal revision). We would appreciate any further suggestion on how to improve PREDICT's description.
>
> > It would be helpful to formally define each of the evaluation metrics
>
> Thank you for raising this. We have provided a formal definition for each of the evaluation metrics on page 14 of the paper update (via rebuttal revision) and will incorporate them into the paper' main body.
>
> > Table 1: how significant are there results as most of the differences fall within the std margin. How would Table 1 support the main claim (Ln 22) that 'PREDICT more accurately infers nuanced human preferences', especially when the Ln 420 concluded that 'PPCM does not show a clear trend' when comparing PREDICT to PREDICT_{CP}
>
> For the standard deviation margins, there is overlap between the ablations and PREDICT full, but not between the ablations and `PREDICT+ICL` nor between `PREDICT_{full}`/`PREDICT+ICL` and the baselines. The ablations that overlap with `PREDICT_{full}` vary based on task. Our conclusion is different components are maximally beneficial for different tasks, preference types, and data types. However, across all tasks, `PREDICT_{full}` outperforms all ablations.
>
> The similarity in performance for PREDICT and PREDICT_{CP} suggests that compound preferences are not necessary for PREDICT to perform well. However, compound preferences can have benefits in scenarios not explored in the paper, such as user interpretability and preference reusability.
>
> We will include this more nuanced discussion in the paper.
>
> > The paper heavily references Gao et al. 2024 for experiment setups, methods, etc without specifying the necessary details within the paper
>
> We include a summary of the paper’s method and benchmark in lines 255 - 266 (PRELUDE in Section 4.3). Are there additional details you would like to see mentioned in the paper? We will add them to Section 4.3.
>
> **Questions**
> > Ln 142 "base components": are they all hand crafted for each task?
>
> PREDICT does not use any handcrafted features or components. Instead PREDICT instructs an LLM to infer the base components or features using the prompt shown in our updated Figure 2. Example base components are also provided in the updated Figure 2. The ground truth preferences are hand constructed by us to ensure that the different preference components induce different demonstrations and that the human proxy can detect their presence. Further details on criteria used to choose the ground truth preferences are in Section 4.3 “Preference Sets”.
>
> > Ln 156, 173, 260: who are the 'users'? Human? LLM proxy?
>
> Here we use an LLM proxy for human users to ensure we know the ground truth preferences, to ensure the “human” demonstrations and evaluation of assistant solutions rely on the same preferences, and for reproducibility of our results and the usefulness of the PLUME benchmark. We will also add this information into the experimental set up section.
>
> > Ln 233: it is unclear what the relevance/importance it is for the environment to be partially observable
>
> The importance is that most real world tasks will involve some degree of partial observability. For example, the user’s mood may not be observable, but can influence the chattiness of their writing. Therefore, we make sure this is represented in our benchmark.

---

> ### Author Response · Authors · 2024-11-21
> **Author Response Follow Up**
>
> Hello,
>
> As we are over halfway through the discussion period, we are reaching out to check in on whether our responses have addressed your concerns. Please let us know if we have not, and what we can do to better address your concerns.
>
> Thank you.

---

> ### Author Response · Authors · 2024-11-25
>
> Hello,
> We have less than two days left in the discussion period.
>
> Based on feedback from reviewers Cces and tU2z, we have simplified our main figure. Now an example is provided only for the writing task in the main body of the paper, and the example for PICK UP is moved to Appendix A.
>
> Please let us know if this updated version and our responses above have not addressed your concerns.
>
> Thank you.

---

> > ### Comment · Reviewer_27u7 · 2024-12-03
> > **Respond to Authors' Comments**
> >
> > Hello! Thanks for your detailed responses and edits. I've raised the score. Best of luck!

---

### Official Review · Reviewer_tU2z · 2024-11-05

**Soundness:** 2
**Presentation:** 3
**Contribution:** 2
**Rating:** 5
**Confidence:** 3

**Summary:**

The paper introduces a method called PREDICT which is designed to infer individual user preferences from user interaction trajectories. The proposed approach is evaluated in two settings: a toy gridworld environment designed by the authors in which an agent must pick up objects on a 5x5 grid while maximizing a reward with user preferences unbeknownst to the agent, and PLUME, an assistive writing preference environment developed by the authors by modifying the recently proposed PRELUDE environment.

Update: The discussion has been helpful in improving the clarity and positioning of the work. I have increased the presentation score and overall rating.

**Strengths:**

The proposed idea is quite interesting. The idea of examining counterfactual trajectories seems novel and is intuitive. Additionally, it appears the proposed PLUME environment addresses valid limitations of prior work. The presented main experimental section also includes several important ablations to consider, and the main results indicate strong improvements over the presented baselines.

**Weaknesses:**

*The writing is somewhat confusing.* For instance, Figure 2 is pretty underspecified and hard to parse, and many of the important details are not available in the main text (e.g., in L122-139 it is hard to concretely understand what the algorithm is doing; preference aggregation in L165-169 is unclear).

*The overall interaction setting seems unrealistic.* The PICKUP environment is rather toy-ish, and it is difficult to say whether such types of preference identification (determining what colors a user prefers) is actually reflective of real-world personalization settings (e.g., linguistic style, interaction length). While PLUME appears to address some of these concerns, it is actually not clear to me whether the cost function is representative of user preference. The number of users seems quite small (only five for summarization and four for email writing).

*The baseline comparisons are rather limited.* In Figure 3, PREDICT may outperform CIPHER-1, but it underperforms ICL for all LLMs evaluated. While it seems that PREDICT + ICL outperforms ICL alone in Table 1, this raises questions about comparing PREDICT with more sophisticated prompt-based baselines which may involve examining trajectories and requiring similar levels of complexity as PREDICT + ICL (for instance, Monte-Carlo Tree Search). Relatedly, Behavior Cloning is the only baseline for the gridworld environment whereas it may be more expected to compare against standard RL-based baselines in such environments. While the authors also claim that PREDICT is not limited by the issues of fine-tuning, it is not clear to me that it is necessarily advantageous compared to tuning.

**Questions:**

What setting would you propose in which such learned preferences are actionable?

How do you obtain sufficient user trajectories in practice?

---

> ### Author Response · Authors · 2024-11-15
> **Author Response**
>
> **Weaknesses**
> > The writing is somewhat confusing. For instance, Figure 2 is pretty underspecified and hard to parse, and many of the important details are not available in the main text (e.g., in L122-139 it is hard to concretely understand what the algorithm is doing; preference aggregation in L165-169 is unclear).
>
> Thank you for pointing this out. Other reviewers raised similar concerns and with hindsight we agree it is currently difficult to concretely understand the algorithm. To improve the description of the method, we updated Figure 2 (now Figure 1 in the paper update via rebuttal revision) to contain more details and rewritten the Iterative Refinement section (now pages 3-4 in the paper update via rebuttal revision). We would appreciate any further suggestion on how to improve PREDICT's description.
>
> > The overall interaction setting seems unrealistic. The PICKUP environment is rather toy-ish, and it is difficult to say whether such types of preference identification (determining what colors a user prefers) is actually reflective of real-world personalization settings (e.g., linguistic style, interaction length). While PLUME appears to address some of these concerns, it is actually not clear to me whether the cost function is representative of user preference. The number of users seems quite small (only five for summarization and four for email writing).
>
> The overarching interaction setting can take one of two forms: (1) the assistant is able to observe a user as the user completes daily tasks; or (2) the user provides demonstrations for specific tasks the user would like the assistant to take over.
>
> While the PICKUP domain does not have a direct correlate to a real world task, mapping object attributes (e.g. color and shape) to preferences does have correlates with real world tasks. We include PICKUP to show that PREDICT can be applied to non-language based tasks where preferences are over factors other than linguistic style or interaction length. PICKUP is a baseline environment where we can specify different user preferences and ensure that policies serve as human proxies. Additionally, the preferences need to be discrete so the quality/accuracy of the inferred preferences can be validated in a straightforward and reliable manner.
>
> Regarding PLUME and whether the cost function is representative of user preference, PLUME does not use a cost function. PRELUDE, the benchmark we extend, uses the Levenshtein distance between the assistant’s solution and the human proxy’s rewrite of the assistant’s solution as a cost function. In PLUME, we generalize cost functions to evaluation metrics (which allows for “higher is better” or “lower is better” metrics) and move away from Levenshtein distance in favor of a Per Preference-Component Match (PPCM) over concerns about the correlation between the evaluation metric and the inferred preferences. PPCM uses LLM-as-a-Judge on a Likert Scale from -2 to +2, and it achieves a strong Pearson correlation of 0.76 with the quality of inferred preference, demonstrating that it is representative of user preferences. For more details, see section 4.3, subheader Metric Correlation and Appendix C.3. We can see how this confusion may have arisen from the manuscript. To address this confusion, we will update the paper to explicitly state that no cost function is used in PLUME and update the references to “cost” to make it clear we are talking about the performance metrics.
>
> For PLUME there are more than five users for summarization and four for email writing. These are the number of user profiles, but each profile is used to instantiate five different users. The user profile determines the user preferences and the random seed determines how the user expresses those preferences. Therefore, we have 25 user combinations for summarization and 20 for email writing. We will update the manuscript to make sure this is clearly stated. The number of user profiles and their source documents are based on prior work, specifically the PRELUDE benchmark [Gao et al., 2024].

---

> ### Author Response · Authors · 2024-11-15
> **Author Response 2**
>
> > The baseline comparisons are rather limited. In Figure 3, PREDICT may outperform CIPHER-1, but it underperforms ICL for all LLMs evaluated. While it seems that PREDICT + ICL outperforms ICL alone in Table 1, this raises questions about comparing PREDICT with more sophisticated prompt-based baselines which may involve examining trajectories and requiring similar levels of complexity as PREDICT + ICL (for instance, Monte-Carlo Tree Search). Relatedly, Behavior Cloning is the only baseline for the gridworld environment whereas it may be more expected to compare against standard RL-based baselines in such environments. While the authors also claim that PREDICT is not limited by the issues of fine-tuning, it is not clear to me that it is necessarily advantageous compared to tuning.
>
> We are unclear what a “more sophisticated prompt-based baseline which may involve examining trajectories” for PLUME would be. Can you please point us to a paper for the type of method you are recommending?
>
> For our ICL baseline, we note that perfect retrieval is used to create the in-context examples, resulting in an upper-bound for how well ICL would be able to perform. Any ICL solution implemented in practice would be unlikely to guarantee the “best” examples were retrieved.
>
> For PICKUP, we do believe BC is the most fair comparison. Note that the reward functions are used for evaluation purposes only.  The reward functions are **not** accessible to the assistant, which is the crux of the problem being addressed, and why all baselines learn from demonstrations. Similarly, we do not consider a PbRL baseline, since our constraint is that the user provides only demonstrations and no other feedback. We will update the evaluation section to make sure these constraints are clear
>
> **Questions**
> > What setting would you propose in which such learned preferences are actionable?
>
> Can you elaborate on what you mean by “actionable?”
>
> By actionable learned preferences, we mean inferred preferences that can be used to steer the assistant's behaviour/generations. For instance, in one PLUME roll-out, an actionable preference that PREDICT learnt was to write in the form of a haiku, which the led the assistant to produce this summary:
> Sausage snug in bun,
> Steamed, spiced, and sauced so splendid
> Street snack spreads swiftly.
>
>
> > How do you obtain sufficient user trajectories in practice?
> Though deployment of PREDICT is outside of the scope of this paper, we could obtain user examples by observing user behavior. If an assistive agent is embedded in a phone or computer’s operating system, then with the user’s permission it could observe users complete a variety of tasks, including the types of writing-based tasks we use in PLUME.

---

> > ### Comment · Reviewer_tU2z · 2024-11-22
> >
> > Thanks for the detailed response!
> >
> > >we updated Figure 2 (now Figure 1 in the paper update via rebuttal revision) to contain more details and rewritten the Iterative Refinement section (now pages 3-4 in the paper update via rebuttal revision)
> >
> > I can appreciate that you spent a lot of effort reorganizing your Figures and adding much more detail to make [new] Figure 1 more informative. However, I think visually speaking, it is now overly packed with text, making it quite difficult to read..
> >
> > >While the PICKUP domain does not have a direct correlate to a real world task, mapping object attributes (e.g. color and shape) to preferences does have correlates with real world tasks. We include PICKUP to show that PREDICT can be applied to non-language based tasks where preferences are over factors other than linguistic style or interaction length. PICKUP is a baseline environment where we can specify different user preferences and ensure that policies serve as human proxies. Additionally, the preferences need to be discrete so the quality/accuracy of the inferred preferences can be validated in a straightforward and reliable manner.
> >
> > Thank you for clarifying this point. I still am not quite convinced about the experiments with PICKUP. Even if it is a "baseline environment" I believe it is still too unrealistic to represent real user preferences and the grid environment is relatively low-dimensional. I would be more convinced if you can demonstrate your approach can generalize to a more realistic and complex task.
> >
> > >For PLUME there are more than five users for summarization and four for email writing. These are the number of user profiles, but each profile is used to instantiate five different users.  ...
> >
> > Thank you for clarifying this as well. Could you please demonstrate some examples regarding the diversity captured by the 25/20 users?
> >
> > >more sophisticated baseline which may involve examining trajectories
> >
> > As I mentioned in my review Monte-Carlo Tree Search comes to mind when considering trajectory simulation to learn preferences. I am willing to be wrong about why these types of baselines do not apply to your particular setting, but if that is the case, it would be helpful to have discussion about how your task inputs are incompatible with existing trajectory simulation approaches
> >
> > [1] Don't throw away your value model! Generating more preferable text with Value-Guided Monte-Carlo Tree Search decoding, COLM 2024
> >
> > [2] Monte Carlo Tree Search Boosts Reasoning via Iterative Preference Learning
> >
> > >perfect retrieval is used to create the in-context examples, resulting in an upper-bound for how well ICL would be able to perform
> >
> > Even if you already have identified an optimal set of retrieved examples, is it not possible that having additional examples would form a stronger inductive bias to help guide the task? e.g., [3] talks about how the structure of examples in ICL may be more important than content; [4] talks about how many-shot ICL can match tuning performance
> >
> > [3] Rethinking the Role of Demonstrations: What Makes In-Context Learning Work?, EMNLP 2022
> >
> > [4] Many-Shot In-Context Learning, NeurIPS 2024
> >
> > > For instance, in one PLUME roll-out, an actionable preference that PREDICT learnt was to write in the form of a haiku, which the led the assistant to produce this summary: Sausage snug in bun, Steamed, spiced, and sauced so splendid Street snack spreads swiftly.
> >
> > Thank you for your helpful clarification. In this case, would it not be a reasonable baseline to separately train an agent to identify the preferences separately from the task model which produces the summary?
> >
> > ---
> >
> > Again, thanks for your detailed reply. I am willing to be wrong on any potential misunderstandings here, but I think these are important factors to consider.

---

> > > ### Author Response · Authors · 2024-11-22
> > > **Author Response 1**
> > >
> > > Thank you for your clarification and comments! Please see our responses below.
> > >
> > > > I can appreciate that you spent a lot of effort reorganizing your Figures and adding much more detail to make [new] Figure 1 more informative. However, I think visually speaking, it is now overly packed with text, making it quite difficult to read..
> > >
> > > Thank you for your feedback. We will revise the figure again to try and find a balance between the two. In your opinion, what would be more valuable to include in the figure: (1) a figure that shows all the details, but only for PLUME (i.e. use the top half of the figure), or (2) showing less detail, but keep both environments (i.e. remove the specific examples and only keep the flow of information)?
> > >
> > > > Thank you for clarifying this point. I still am not quite convinced about the experiments with PICKUP. Even if it is a "baseline environment" I believe it is still too unrealistic to represent real user preferences and the grid environment is relatively low-dimensional. I would be more convinced if you can demonstrate your approach can generalize to a more realistic and complex task.
> > >
> > > We consider the PICKUP task to be supplementary to the PLUME email and summarization tasks to allow us to understand exactly how well PREDICT is able to infer “user” preferences. It also allows us to evaluate PREDICT on a task that is not strictly in-distribution for the LLM used to infer preferences. Prior work (Gao et al., 2024) has relied solely on the email and summarization tasks.
> > >
> > > There is a lack of non-writing-based, interactive tasks that allow for specifying preferences for how a given task should be completed and then steering agents (our proxy human) to behave in accordance with those preferences. If you have an environment you recommend, that would be immensely helpful.
> > >
> > > > As I mentioned in my review Monte-Carlo Tree Search comes to mind when considering trajectory simulation to learn preferences. I am willing to be wrong about why these types of baselines do not apply to your particular setting, but if that is the case, it would be helpful to have discussion about how your task inputs are incompatible with existing trajectory simulation approaches
> > > [1] Don't throw away your value model! Generating more preferable text with Value-Guided Monte-Carlo Tree Search decoding, COLM 2024
> > > [2] Monte Carlo Tree Search Boosts Reasoning via Iterative Preference Learning
> > > Thank you for highlighting these papers, as they are complementary with PREDICT and highlight additional use cases for the inferred preferences. These methods may additionally be compatible with ICL, however it would require novel components to score nodes based on the contexts, and therefore should not be considered a baseline.
> > >
> > > It looks like [1] was published in COLM 2024, which occurred Oct. 7 -9 after the Oct. 1 ICLR deadline. [2] has yet to be presented at a peer-reviewed conference, potentially rejected from ACL 2024. As such both of these papers should be considered contemporary work per the ICLR guidelines: “...if a paper was published (i.e., at a peer-reviewed venue) on or after July 1, 2024, authors are not required to compare their own work to that paper. Authors are encouraged to cite and discuss all relevant papers, but they may be excused for not knowing about papers not published in peer-reviewed conference proceedings or journals, which includes papers exclusively available on arXiv.” (https://iclr.cc/Conferences/2025/ReviewerGuide). We are happy to discuss both work as contemporary and what they mean for PREDICT.
> > >
> > > Having read the two papers, neither learn preferences nor try to imitate a user, and as such cannot be directly applied to the problem that PREDICT tackles. Both improve an LLMs ability to decode language to maximize a provided preference. In [1], the LLM uses MCTS to steer sentiments, reduce toxicity, be introspective on common sense reasoning, and improve harmfulness/helpfulness. In [2], they use MCTS for RLAIF, where an LLM-as-a-Judge guides the decoding to create training data that it itself determines as more valuable. In other words, both works utilize MCTS to generate language that is more aligned with a provided directive, but neither attempt to identify what the directive is. In contrast, PREDICT directly identifies the directive (e.g. the user’s preference).

---

> > > ### Author Response · Authors · 2024-11-22
> > > **Author Response 2**
> > >
> > > > Even if you already have identified an optimal set of retrieved examples, is it not possible that having additional examples would form a stronger inductive bias to help guide the task? e.g., [3] talks about how the structure of examples in ICL may be more important than content; [4] talks about how many-shot ICL can match tuning performance
> > > [3] Rethinking the Role of Demonstrations: What Makes In-Context Learning Work?, EMNLP 2022
> > > [4] Many-Shot In-Context Learning, NeurIPS 2024
> > >
> > > Additional examples and curated examples may indeed improve the performance of ICL, however the same is equally true of PREDICT. In our setting, PREDICT and ICL are provided with identical, high-performing retrieval mechanisms and the same number of samples to learn from, enabling a fair comparison.
> > >
> > >
> > > > Thank you for clarifying this as well. Could you please demonstrate some examples regarding the diversity captured by the 25/20 users?
> > >
> > >
> > > Below is a pair of examples written by different users that share the following preferences:
> > > - include a simile
> > > - adopt a step-by-step structure
> > > - use ampersands (&) instead of "and"s
> > > - write in the style of a children's book
> > >
> > > However, because they are summarizing different articles, the way that the preferences are exhibited are slightly different (for example “Step 1:” vs “1.” and the inclusion of the “The end.” in the second example).
> > >
> > > User 1
> > > Once upon a time, there was a cricketer named Chris Woakes who was like a hidden gem waiting to be discovered. He wasn't always the star of the show, but that was about to change!
> > >
> > > Step 1: Chris Woakes played in Sri Lanka & took his chance to shine. He bowled so well that everyone started to notice him.
> > >
> > > Step 2: In a big match against Australia, Chris was given the new ball to bowl with. He bowled fast & took lots of wickets, even though England didn't win the game.
> > >
> > > Step 3: Chris felt more confident now. He said, "It's nice to know I've performed well at the international level." He was happy to be a part of the team.
> > >
> > > Step 4: Chris traveled to Brisbane with his team to play against India. He remembered how he had taken six wickets in a game there four years ago. He wanted to do it again!
> > >
> > > Step 5: Chris worked very hard on his bowling. He practiced with his coaches & improved his speed. He was now bowling faster than ever before!
> > >
> > > Step 6: Chris was not just a bowler. He could also hit the ball hard when batting & was a great fielder. One time, he even juggled a catch with his feet before holding onto it!
> > >
> > > Step 7: Chris's hard work paid off. He became an important player for England. With other bowlers like Jimmy Anderson & Stuart Broad recovering from injuries, Chris's role was crucial.
> > >
> > > And so, Chris Woakes, the once-hidden gem, became a shining star for England's cricket team, ready to take on the world!
> > >
> > >
> > > User 2
> > > Once upon a time, scientists thought only big, gassy planets like Saturn could wear rings, like a king with a shiny crown. But guess what? They were wrong! Here's the story, step-by-step:
> > >
> > > 1. Far, far away, between Saturn & Uranus, there’s a rocky asteroid named Chariklo.
> > > 2. Chariklo was once part of the Kuiper Belt, a magical place full of icy dwarf planets & comets at the edge of our solar system.
> > > 3. One day, Chariklo got tossed out of the Kuiper Belt & became the biggest member of a group called the Centaurs.
> > > 4. Scientists, using their powerful telescopes, were watching Chariklo closely. They weren’t looking for anything special, just curious about this space rock.
> > > 5. To their surprise, they discovered that Chariklo had not one, but TWO rings! Just like Saturn, but much smaller.
> > > 6. These rings are made of ice particles & pebbles, & they are very thin—only two & five miles wide.
> > > 7. The rings are separated by a gap of five miles, & they are only a few hundred meters thick.
> > > 8. This amazing discovery was so precise that scientists could measure the rings from two billion kilometers away!
> > >
> > > And that’s how Chariklo, the little asteroid, surprised everyone by wearing rings just like a planet. The end.

---

> > > ### Author Response · Authors · 2024-11-25
> > >
> > > Thank you for feedback on our main figure. Based on your feedback and that of reviewer Cces, we have simplified the figure. Now an example is provided only for the writing task and the example for PICK UP is moved to Appendix A.
> > >
> > > Please let us know if this updated version and our responses above have not addressed your concerns.

---

> > > > ### Comment · Reviewer_tU2z · 2024-11-25
> > > >
> > > > Thank you for your detailed response. I think the revisions to the paper are great progress and I appreciate the effort that has been put into the rebuttal.
> > > >
> > > > Re: Figure 1
> > > >
> > > > I think the reduction to only one task has helped a lot. I think in general, though, you do not need to provide the full-length text examples in the Figure. For example,
> > > >
> > > > "A hot dog is a sausage. It can be topped with various condiments like mustard. Hot dogs became popular in the U.S" -> "A hot dog is a sausage. It can be ..."
> > > >
> > > > These types of changes may help reduce the visual clutter? But if you feel that the full-length text is necessary feel free to ignore my suggestion.
> > > >
> > > > Re: PICKUP & Baselines
> > > >
> > > > My primary concerns are still concerning the unrealistic nature of PICKUP, and the choice of weak baselines. It would be helpful to see that PREDICT leads to learned user preference, and that it leads to improved interaction outcomes. For instance,   I believe that the planning environment in [1] can be objectively evaluated with respect to improved goal outcomes, and a higher reward indicates itinerary selections which better represent user preference. [2] also uses MCTS to plan pragmatic-level actions, with the idea that an improved action sequence will lead to improved user persuasion. From my perspective, higher is persuasion rates are also indicative of improved understanding of an individual user's preferences. My feeling is that both of these are real-world tasks with non-linguistic preferences. Although preference may not be formulated with the exact same problem setup, I think that there is a reasonable mapping between the two task setups.
> > > >
> > > > [1] Decision-Oriented Dialogue for Human–AI Collaboration
> > > >
> > > > [2] Prompt-Based Monte-Carlo Tree Search for Goal-Oriented Dialogue Policy Planning
> > > >
> > > > Re: ICL baseline and examples
> > > >
> > > > Thank you for clarifying and providing these examples, they are quite helpful.
> > > >
> > > > ---
> > > >
> > > > Overall, given our discussion and the other reviewers' comments, I am willing to increase my overall score. The author discussion has helped quite a lot. However, the biggest questions from my perspective are still regarding the evaluation tasks in the current version of the paper, and the baseline methods compared.

---

> ### Author Response · Authors · 2024-11-21
> **Author Response Follow Up**
>
> Hello,
>
> As we are over halfway through the discussion period, we are reaching out to check in on whether our responses have addressed your concerns. Please let us know if we have not, and what we can do to better address your concerns.
>
> Thank you.

---

> ### Author Response · Authors · 2024-11-27
> **Author Response**
>
> Thank you for your engagement and the discussion.
>
> We understand your desire for additional baselines. However, per ICLR policy in contemporary work, the two you referenced are out of bounds for this IClR submission due to either not being part of a peer reviewed proceedings or having been part of COLM2024, which occurred after the Oct. 1 ICLR deadline. Therefore, additional baselines beyond what we used are not clear nor available.
>
> For the additional environments you have referenced, a key characteristic of PREDICT is to learn linguistically expressible preferences. Therefore, the two environments relying on "non-linguistic preferences" makes them a poor match for PREDICT.

---

> > ### Comment · Reviewer_tU2z · 2024-12-03
> >
> > Thanks again for continuing to engage in this discussion. As I have mentioned in my last response, I have increased my overall score as the discussion has helped. But to respond to your points in your modified response:
> >
> > > For the additional environments you have referenced, a key characteristic of PREDICT is to learn linguistically expressible preferences. Therefore, the two environments relying on "non-linguistic preferences" makes them a poor match for PREDICT.
> >
> > I suggested the two environments based on your comment in your response: "We include PICKUP to show that PREDICT can be applied to non-language based tasks where preferences are over factors other than linguistic style or interaction length." I had mentioned them because they are "non-language based" preferences, but the action choices themselves are "linguistically expressible" (i.e., dialogue actions). Additionally, in the Decision-Oriented Dialogue paper, the planning task is to "propose possible multi-day itineraries based on partial knowledge of the user’s preferences and domain knowledge, and iteratively refine the plan with the user" so you could also say that the user's preferences are linguistically expressed.
> >
> > > However, per ICLR policy in contemporary work, the two you referenced are out of bounds for this IClR submission due to either not being part of a peer reviewed proceedings or having been part of COLM2024, which occurred after the Oct. 1 ICLR deadline. Therefore, additional baselines beyond what we used are not clear nor available.
> >
> > The Value-Guided Monte-Carlo Tree Search paper has been on arXiv since September 2023 and was publicly accepted at COLM in July of this year. While this may be a gray area, I gave Monte-Carlo Tree Search in my initial review as a broad example. The other paper I mentioned when discussing environments ("Prompt-Based Monte-Carlo Tree Search for Goal-Oriented Dialogue Policy Planning") was published at EMNLP 2023, and a quick search on MCTS yields many other recent works which have received much attention when applied to planning, reasoning, etc. I would be more convinced that "additional baselines are not clear nor available" if you can demonstrate that it is actually not possible to adapt such baselines to your task.

---

### Meta-Review · Area_Chair_wSU6 · 2024-12-21

**Metareview:**

This paper introduces PREDICT, a method that enhances the precision and adaptability of inferring human preferences in AI interactions by refining preferences iteratively, decomposing them into components, and validating them across multiple trajectories. Evaluated in both a gridworld and a new text-domain environment, PREDICT demonstrates significant improvements over existing baselines. Despite its contributions, according to the reviewers' feedback and discussions, this paper still has to be improved on the following aspects: unrealistic interaction settings, limited baseline comparisons, lack of detailed examples and formal definitions, insufficient investigation of iterative refinement steps, reliance on synthetic users without real case analyses, and the presentation of the methodologies.

**Additional Comments On Reviewer Discussion:**

The discussions are thorough and in multi-turns.

---

### Decision · Program_Chairs · 2025-01-22

Reject